# Orexin modulates behavioral fear expression through the locus coeruleus

Shingo Soya[1], Tohru M. Takahashi[1,2], Thomas J. McHugh [3], Takashi Maejima [2], Stefan Herlitze[4], Manabu Abe[5], Kenji Sakimura[5] & Takeshi Sakurai[1,6,7]

Emotionally salient information activates orexin neurons in the lateral hypothalamus, leading to increase in sympathetic outflow and vigilance level. How this circuit alters animals' behavior remains unknown. Here we report that noradrenergic neurons in the locus coeruleus (NA[LC] neurons) projecting to the lateral amygdala (LA) receive synaptic input from orexin neurons. Pharmacogenetic/optogenetic silencing of this circuit as well as acute blockade of the orexin receptor-1 (OX1R) decreases conditioned fear responses. In contrast, optogenetic stimulation of this circuit potentiates freezing behavior against a similar but distinct context or cue. Increase of orexinergic tone by fasting also potentiates freezing behavior and LA activity, which are blocked by pharmacological blockade of OX1R in the LC. These findings demonstrate the circuit involving orexin, NA[LC] and LA neurons mediates fear-related behavior and suggests inappropriate excitation of this pathway may cause fear generalization sometimes seen in psychiatric disorders, such as PTSD.

[1] International Institute for Integrative Sleep Medicine (WPI-IIIS), University of Tsukuba, 1-1-1 Tennodai Tsukuba, Ibaraki 305-8575, Japan. [2] Department of Molecular Neuroscience and Integrative Physiology, Faculty of Medicine, Kanazawa University, Kanazawa, Ishikawa 920-8640, Japan. [3] Laboratory for Circuit & Behavioral Physiology RIKEN Brain Science Institute, Wako, Saitama 351-0198, Japan. [4] Department of General Zoology and Neurobiology, ND7/31, Ruhr-University Bochum, Universitätsstrasse 150, 44780 Bochum, Germany. [5] Department of Cellular Neurobiology, Brain Research Institute, Niigata University, Asahimachi, Chuoku Niigata 951-8585, Japan. [6] Faculty of Medicine, University of Tsukuba, Tsukuba, Ibaraki 305-8575, Japan. [7] Life Science Center for Tsukuba Advanced Research Alliance (TARA), University of Tsukuba, 1-1-1 Tennodai Tsukuba, Ibaraki 305-8575, Japan. Correspondence and requests for materials should be addressed to T.S. (email: sakurai.takeshi.gf@u.tsukuba.ac.jp)

Learned fear is an adaptive behavioral response that allows animals to predict potential threats based on memory of past aversive experiences. While there are behavioral advantages to the association of aversive experiences previously paired with a distinct stimulus or context to related cues or similar situations, a process known as fear generalization, this ability must be appropriately controlled to be beneficial for an organism. The classical Pavlovian fear conditioning model is well suited to understanding how animals learn stimuli in a particular environment signal threats, and the neuronal mechanisms underlying this function have been extensively studied[1,2]. However, the neuronal mechanisms as to how fear learning is generalized or modulated to adapt other stimuli that contain similar, but distinct elements of a learned threat remain largely unexplored. Here, we found that orexin plays a role in regulation of the level of fear-related responses in situations that resemble a past aversive experience.

**Fig. 1** Role of OX1R in NA[LC] neurons in cued fear memory retrieval. **a** Schematic drawing of fear conditioning paradigm used in this study. **b** Cell-type-selective deletion of OX1R in NA[LC] neurons. *LoxP* sites are introduced in the *OX1R* allele in *OX1R^{f/f}* mice to delete exons 5 and 6 by Cre-mediated recombination. **c**, **d** After cued fear conditioning, *OX1R^{f/f};NAT-Cre* mice reduced freezing responses in test session. **e** Pharmacological blockade of OX1R by OX1R antagonist (SB334867) in cued fear conditioning paradigm. **f**, **g** After cued fear conditioning, i.p. administration of SB334867 1 h before the test session significantly attenuated cued fear memory retrieval. **p < 0.01, ***p < 0.001. Values are mean ± SEM

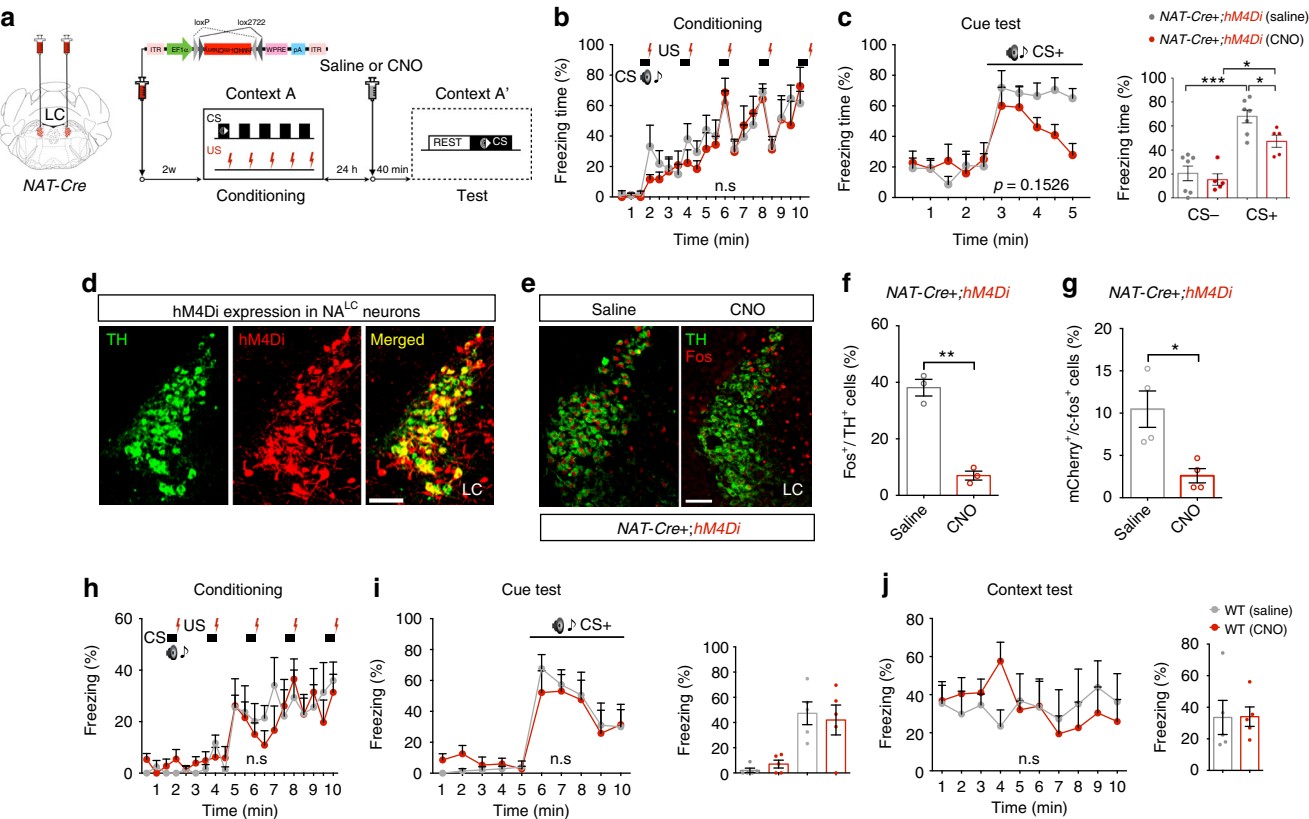

**Fig. 2** Pharmacogenetic inhibition of NA$^{LC}$ neurons decreased cue-induced freezing behavior. **a** Experimental protocol. **b, c** After cued fear conditioning, CNO (clozapine-N-oxide) administration 40 min before the test session decreased freezing behavior against a conditioned stimulus (CS) in test session. **d** Images showing that AAV-mediated hM4Di-mCherry (red) expression is especially observed in TH-positive NA$^{LC}$ neurons (green) of *NAT-Cre* mice. Scale bar, 100 μm. **e** Images showing Fos/TH-double-labeled cells in the LC after test session in saline- or CNO-treated mice. Scale bar, 100 μm. **f** Percentage of Fos$^+$/TH$^+$ cells in the LC. **g** Percentage of Fos$^+$/mCherry$^+$ cells in the LC. **h–j** After cued fear conditioning, CNO does not affect cued and contextual fear expression in wild-type mice. *$p < 0.05$, **$p < 0.01$, ***$p < 0.001$. Values are mean ± SEM

Orexin plays a highly important role in maintenance of arousal[3,4]. Many studies have suggested that orexin neurons are activated during the behavioral expression of fear or in response to cues associated with danger or reward[4]. Thus, we hypothesized that this activation might play a role in modulating animal's appropriate behavior depending on the situation, because orexin is a key molecule in the shift to a "vigilant" state in emotionally arousing situations[4]. For instance, behavioral and stress responses to intruders and the cardiovascular response to air jet stress were shown to be attenuated in orexin-deficient mice[5]. Orexin neurons receive input from the bed nucleus of the stria terminalis (BST), suggesting that the limbic system regulates activity of these cells in response to emotionally salient stimuli[6,7]. We confirmed that orexin neurons were markedly activated by both cued and contextual fear (Supplementary Fig. 1).

We previously reported that noradrenergic neurons in the locus coeruleus (NA$^{LC}$ neurons), which abundantly express orexin receptor-1 (OX1R), are involved in the expression and/or consolidation of cued fear memory via an activation of the lateral amygdala (LA) neurons[8]. These results suggest the potential role of orexin neurons in the recruitment of LA-projecting NA$^{LC}$ (NA$^{LC→LA}$) neurons to control expression of fear-related behavior. However, the role of this pathway on the behavioral expression of fear itself, rather than consolidation of fear memory, has not been previously addressed.

NA$^{LC}$ neurons have divergent input/output architecture and are involved in various physiological functions such as stress response, attention, arousal, cognitive function, and reward-seeking behavior[9–11]. In addition, NA transmission is involved in anxiety and phobia[12]. In the rodent LA, noradrenaline is known to be a critical modulator of fear memory formation via-adrenergic receptor activation (βARs)[13,14], while in humans, β-AR blockers disrupt fear memory consolidation[15].

While it had been suggested that the lateral hypothalamus (LH), where orexin neurons are localized, is not involved in Pavlovian fear learning[16], recent data demonstrate that orexin neuron plays an important role in fear memory learning and consolidation[8,17], as well as fear memory extinction[18]. From these findings, we hypothesized that orexin neurons, which respond to emotional stimuli, can also modulate the expression of fear via noradrenergic signaling governed by NA$^{LC→LA}$ neurons.

Here, we employed the cell-type-specific tracing of the relationship between input and output (cTRIO) approach[19], to histologically identify monosynaptic connections between orexin neurons and NA$^{LC→LA}$ neurons, and used chemogenetic or optogenetic manipulations of these circuits. Enhancement of orexinergic activity by these manipulations as well as fasting increased fear-related behavioral responses through the OX1R signaling in the LC. This study demonstrates that the pathway constituting orexin$^{LH→LC}$ and NA$^{LC→LA}$ neurons positively regulate and physiologically modulates the level of fear-induced behavior in response to environmental information.

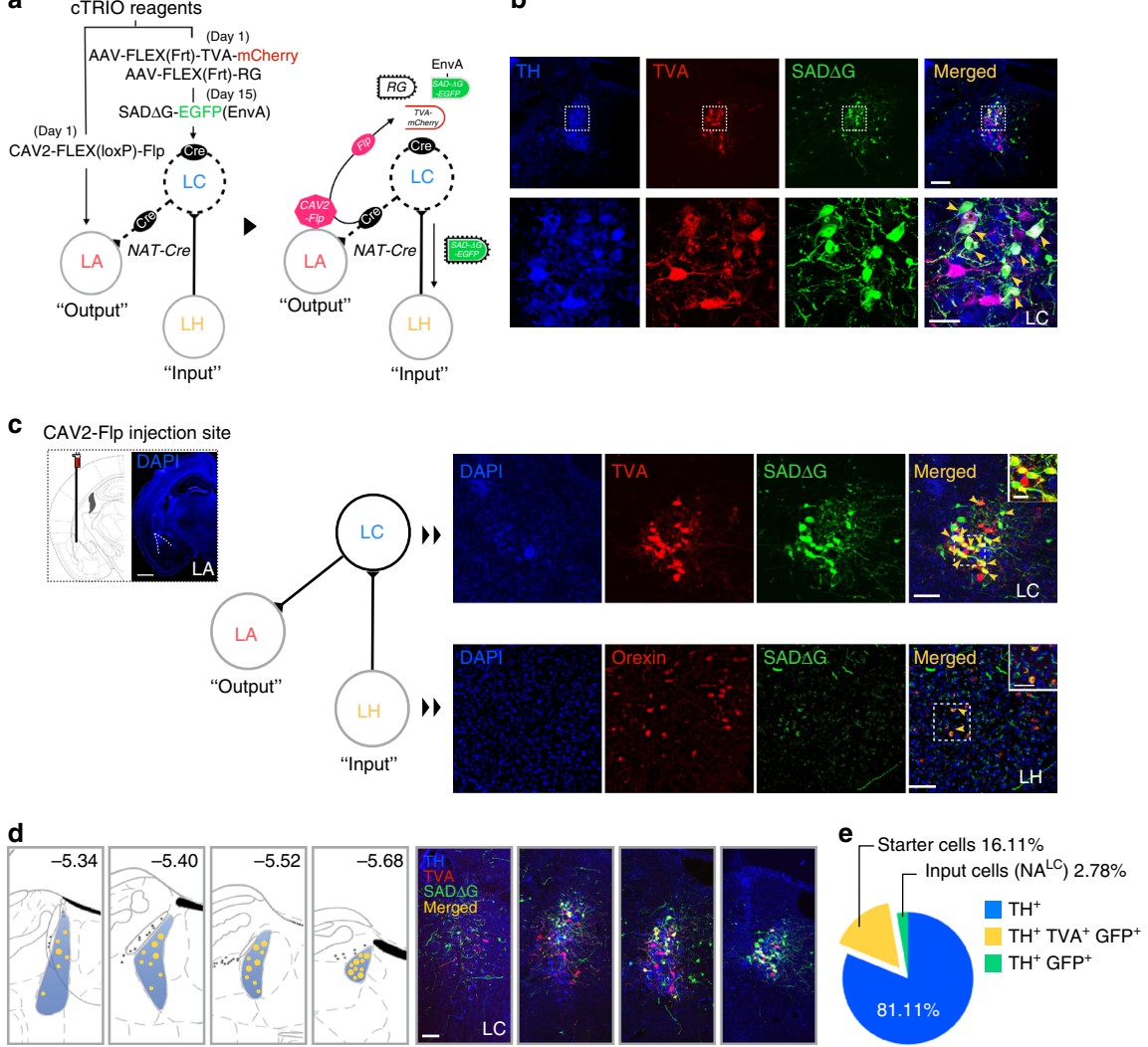

**Fig. 3** Trans-synaptic tracing of orexin$^{LH \to LC}$ to NA$^{LC \to LA}$ circuit by cTRIO. **a** Experimental procedure. We injected *CAV2-FLEX (loxP)-Flp* into the LA of *NAT-Cre* mice to express Flp in LC$^{NA}$ neurons that send projections to the LA (Day 1). Flp-dependent AAV carrying *TVA*, *RG* were then injected into the LC (Day 1). *SADΔG-EGFP (EnvA)* was injected into the same site (Day 15). **b** Upper panels show that the starter cells (TVA$^+$, SADΔG$^+$ cells) in the LC are NA neurons. From left to right, TH$^+$, TVA$^+$, SADΔG-EGFP$^+$, and merged cells (shown by yellow arrow heads) are shown. Lower panels, higher power images of corresponding upper panels of the rectangular regions. Scale bars, 200 µm (upper), 50 µm (lower). **c** Left panels, injection site of *CAV2-FLEX (loxP)-Flp* in the LA. Scale bar, 600 µm. Right upper panels, NA$^{LC}$ neurons expressing TVA and SADΔG-EGFP (yellow arrow heads). Scale bars: Large window, 100 µm; small window, 30 µm. Lower panels, orexin$^{LH}$ neurons expressing SADΔG-EGFP (yellow arrow heads), suggesting that NA$^{LC \to LA}$ neurons receive monosynaptic input from orexin neurons. Scale bars: large window, 100 µm; small window, 50 µm. **d** Left four panels, distribution of starter cells (merged) in the LC region. Each small yellow dot shows one merged cell, large dot includes two merged cells. Right four panels show a representative image of starter cells in different locations of the LC. Scale bar: 100 µm. **e** Graph shows the percentage of TH$^+$ (blue), TH$^+$/TVA$^+$/SADΔG$^+$ (yellow), TH$^+$/SADΔG$^+$ (green) cells in the LC

## Results

**Role of OX1R in NA$^{LC}$ neurons in fear expression.** Mice with a global deletion of the OX1R demonstrated decreased freezing in both cued and contextual fear tests, and rescue of OX1R only in NA$^{LC}$ neurons completely normalized the phenotype[8]. Here, using mice lacking the OX1R receptor specifically in NA$^{LC}$ neurons (*OX1R$^{f/f}$; NAT-Cre* mice) (Supplementary Fig. 2), we confirmed the role of OX1R in NA$^{LC}$ neurons in cued fear conditioning test (Fig. 1a, b). While both genotypes showed similar freezing levels during training (*OX1R$^{f/f}$*, $n = 8$; *OX1R$^{f/f}$*; *NAT-Cre*, $n = 12$: two-way RM ANOVA with Sidak's post-hoc test, $F_{(1, 18)} = 2.673$, $p = 0.1194$, Fig. 1c), *OX1R$^{f/f}$; NAT-Cre* mice showed a significant deficit in freezing in test session (freezing over time: two-way RM ANOVA with Sidak's post-hoc test, $F_{(1, 18)} = 7.177$, $p = 0.0153$, Fig. 1d, left; average freezing: unpaired two-tailed Student's $t$ test, $t = 2.994$, $p = 0.0085$, Fig. 1d, right).

*NAT-Cre* mice with normal *OX1R* alleles did not show any abnormality in the cued- and contextual fear test as compared with wild-type littermates, suggesting that existence of *NAT-Cre* allele did not affect the phenotype (Supplementary Fig. 2f–h). This phenotype of *OX1R$^{f/f}$; NAT-Cre* was highly similar to the findings in *OX1R$^{-/-}$* mice in our previous work[8] and suggests that OX1R in NA$^{LC}$ neurons plays a major role in fear memory consolidation and/or expression. To examine whether the function of OX1R is necessary particularly during the fear memory retrieval period, we next used acute pharmacological blockade with a selective OX1R antagonist (SB334867) (Fig. 1e). One day after fear conditioning, in which freezing was comparable between genotypes (vehicle, $n = 13$; SB334867, $n = 13$, two-way RM with Sidak's post-hoc test, $F_{(1, 24)} = 1.086$, $p = 0.3077$, Fig. 1f), mice receiving an intraperitoneal (i.p.) injection of SB334867 (5 mg/kg)[18] 1 h before the test session. The effect of the

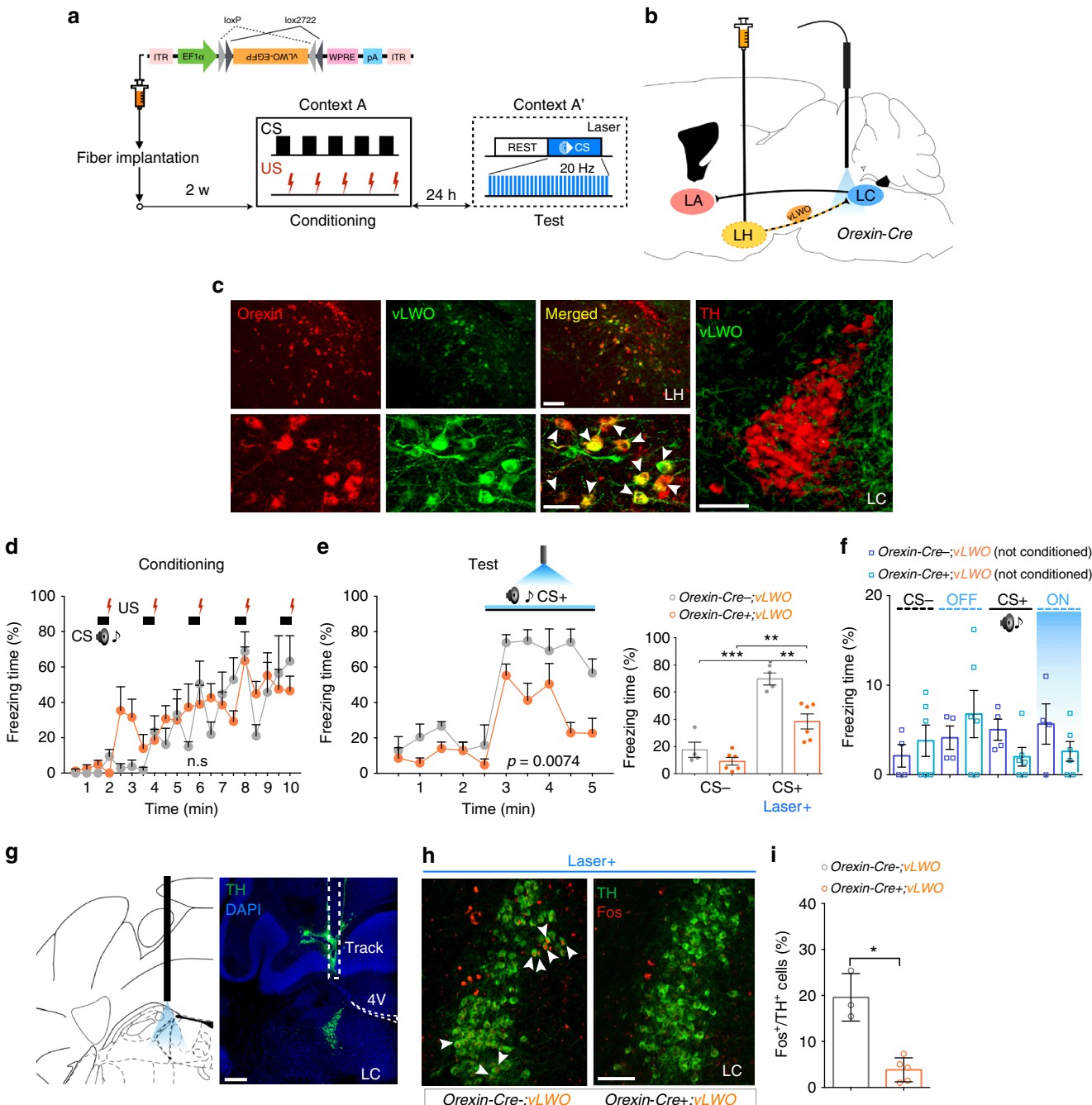

**Fig. 4** Optogenetic inhibition of orexin$^{LH \to LC}$ decreased cued fear expression. **a** Experimental protocol. **b** Schematic representation showing sites for injection of $AAV_{10}$-FLEX-vLWO-EGFP and optic fiber. **c** Histological examination showing that orexin neurons express vLWO. Orexin is stained red, while vLWO-EGFP is stained green. Merged cells were shown by white arrow heads. Scale bars: upper panel, 100 μm; lower panel, 50 μm; right panel, 100 μm. Many vLWO-EGFP-positive fibers are observed in the LC. **d**, **e** After cued fear conditioning, acute optogenetic inhibition reduced freezing behavior against CS. **f** Laser stimulation or auditory CS itself had no effect in naive mice. **g** Track of an optic fiber implanted above the LC. Scale bar, 200 μm. **h** Images showing Fos$^+$/TH$^+$ cells (white arrow heads) in NA$^{LC}$ neurons with or without optogenetic inhibition of the orexin$^{LH \to LC}$ pathway. Scale bar, 100 μm. **i** Quantification of double-positive cells shown in the LC, suggesting that optogenetic inhibition of orexin$^{LH}$ fibers reduced the activity of NA$^{LC}$ neurons. *$p < 0.05$, **$p < 0.01$, ***$p < 0.001$. Values are mean ± SEM

antagonist was not evident for first 30 s of CS presentation, but we observed reduction in freezing behavior for later period (freezing over time: two-way RM ANOVA with Sidak's post-hoc test, $F_{(1, 24)} = 11.47$, $p = 0.0024$, 3.5–5 min in Fig. 1g, left; average freezing: unpaired two-tailed Student's $t$ test, $t = 4.411$, $p = 0.0002$, Fig. 1g, right). These results demonstrate that OX1R activates the NA$^{LC}$ neurons to maintain behavioral fear expression.

**Inhibition of NA$^{LC}$ neurons reduced cue-induced freezing response**. $OX1R^{f/f}$; $NAT$-$Cre$ mice showed decreased freezing during the retrieval period (Fig. 1d) and pharmacological blockade of OX1R impairs sustained expression of freezing behavior against CS (Fig. 1g), suggesting that OX1R in NA$^{LC}$ neurons modulates and maintains fear expression in response to the external fearful stimuli. To further evaluate this hypothesis, we

perfomed chemogenetic inhibition of $NA^{LC}$ neurons during the cued test session. We bilaterally injected $AAV_{10}$-$EF1\alpha$-FLEX (loxP)-hM4Di-mCherry (hM4Di) in the LC of NAT-Cre mice to express hM4Di specifically in $NA^{LC}$ neurons (Fig. 2a). Two weeks after injection, we performed cued fear conditioning, in which we did not observed any difference in freezing between the groups (NAT-Cre+;hM4Di (Saline), n = 7; NAT-Cre+;hM4Di (CNO), n

= 6: two-way RM ANOVA with Sidak's post-hoc test, $F_{(1, 11)}$ = 0.2306, p = 0.6405, Fig. 2b). One day later, we subjected these mice to a test session 40 min after i.p. injection of CNO to inhibit $NA^{LC}$ neurons. The CNO-treated group showed gradually reduced freezing, with the difference most obvious in the later phase of CS presentation (freezing over time: two-way RM ANOVA with Sidak's post-hoc test, $F_{(1, 11)}$ = 5.145, p = 0.1526,

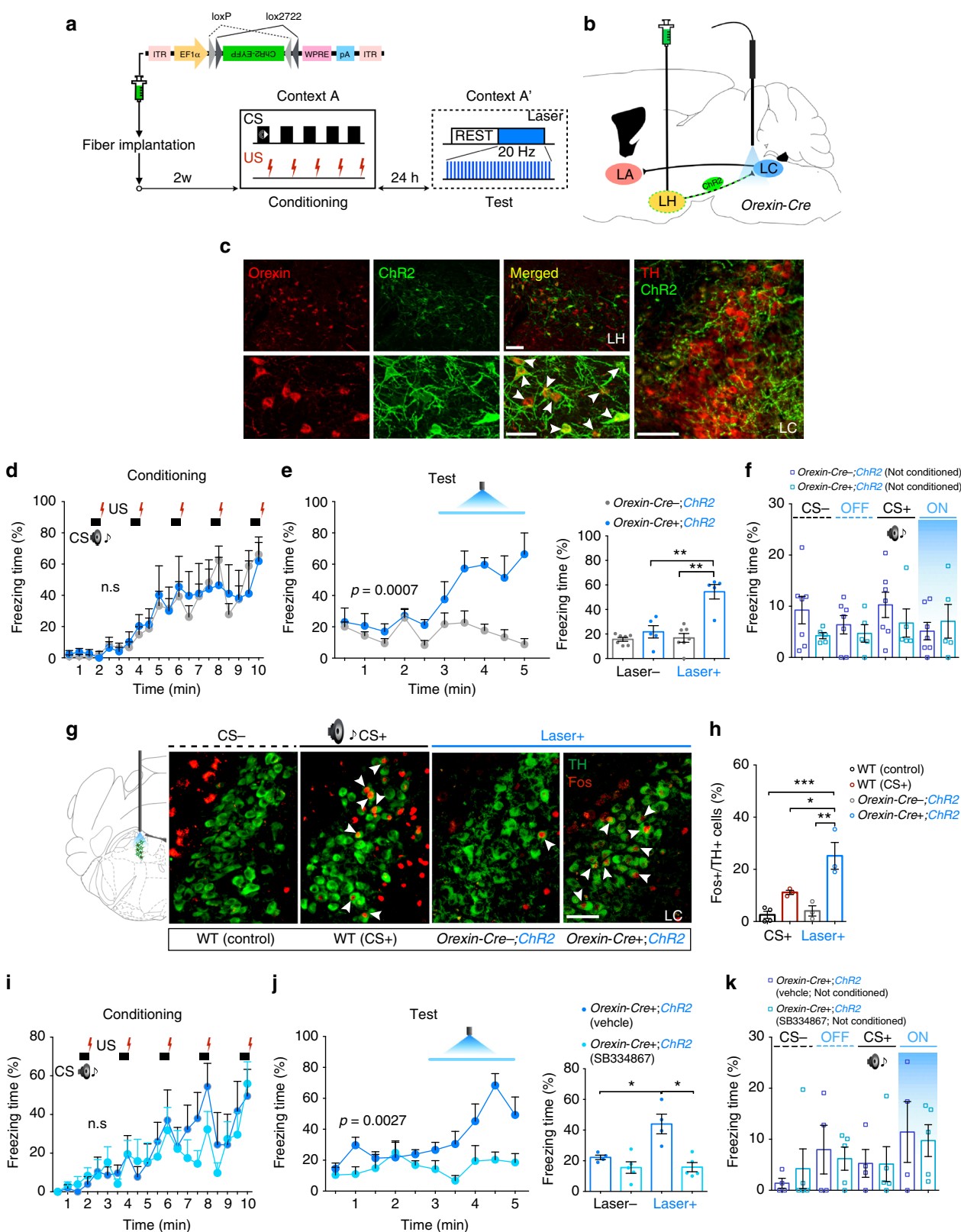

Fig. 2c, left; average freezing: unpaired two-tailed Student's $t$ test, $t = 3.573$, $p = 0.0044$, Fig. 2c, right). After testing, we confirmed specific expression of hM4Di in NA$^{LC}$ neurons (Fig. 2d), and quantified c-Fos expression in TH-labeled cells and Fos- and mCherry-labeled cells in the LC, finding a significant reduction following CNO injection (NAT-Cre+;hM4Di (Saline), $n = 3$; NAT-Cre+;hM4Di (CNO): unpaired two-tailed Student's $t$ test, $n = 3$: $t = 9.309$, $p = 0.0024$, Fig. 2e, f) (NAT-Cre+;hM4Di (Saline), $n = 4$; NAT-Cre+;hM4Di (CNO): unpaired two-tailed Student's $t$ test, $n = 4$: $t = 3.404$, $p = 0.0283$, Fig. 2g). To exclude the possibility that CNO itself affects freezing behavior, we tested the effect of i.p. injection of CNO in wild-type mice. After cued fear conditioning (saline, $n = 5$; CNO, $n = 5$: two-way RM ANOVA, $F_{(1, 8)} = 0.03348$, $p = 0.8594$, Fig. 2h), fear response did not change between saline- and CNO injected group against cued (two-way RM ANOVA with Sidak's post-hoc test, $F_{(1, 8)} = 0.0003$, $p = 0.9847$) and contextual stimuli (two-way RM ANOVA with Sidak's post-hoc test, $F_{(1, 8)} = 0.00145$, $p = 0.9706$) (Fig. 2i, j, left). Average freezing showed no difference among these groups during CS presentation (unpaired two-tailed Student's $t$ test, $t = 0.3564$, $p = 0.7314$) or exposure to the context A (unpaired two-tailed Student's $t$ test, $t = 0.03794$, $p = 0.9706$) (Fig. 2i, j, right).

**Orexin neurons make direct synaptic input to NA$^{LC \to LA}$ neurons.** Although it has been well established that NA$^{LC}$ neurons receive dense projection from orexin neurons[20], and these neurons innervate the LA to modulate amygdala function[14], confirmation of the direct connections between these components has not been achieved. Using the cTRIO method[19], we examined whether orexin neurons make monosynaptic connections to NA$^{LC \to LA}$ neurons. We injected a Cre-dependent CAV2 vector carrying the flippase (Flp) recombinase (CAV2-FLEX (loxP)-Flp) into the LA of NAT-Cre mice[21]. Because CAV2 infects axons and is retrogradely transported to cell bodies, this combination allows us to express Flp specifically in Cre-expressing NA$^{LC \to LA}$ neurons. We then injected the Flp-activatable AAV vectors, AAV$_{10}$-CAG-FLEX (Frt)-TVA-mCherry and AAV$_{10}$-CAG-FLEX (Frt)-RG, into the LC to express TVA and rabies glycoprotein (RG) in these cells. Finally, we injected SADΔG-EGFP (EnvA)[22] in the same regions (LC) to label upstream neurons that make direct synaptic contact with NA$^{LC \to LA}$ neurons. We confirmed that NA$^{LC}$ (TH-positive) neurons express TVA and SADΔG-EGFP, demonstrating that a population of NA$^{LC}$ neurons send axonal projections to the LA (Fig. 3a, b). As expected, we found that some orexin neurons is the LH express SADΔG-EGFP, indicating that orexin neurons have monosynaptic connectivity to NA$^{LC}$ neurons (Fig. 3c) and suggesting that orexin neurons directly regulate NA$^{LC \to LA}$ neurons. In all experiments ($n = 4$), TH, TVA, SADΔG-EGFP co-labeled neurons were restricted to the caudal region of the LC (Fig. 3d), suggesting that the LA receives input mainly from neurons in the posterior part of the LC. The fraction of starter cells represented $16.11 \pm 0.96\%$ of all NA$^{LC}$ neurons (Fig. 3e). We also found input cells in regions other than the LH, including the paraventricular nucleus of the hypothalamus (PVN) and the central nucleus of the amygdala (CeA) and medial

parabrachial nucleus (MPB), suggesting that NA$^{LC \to LA}$ neurons receive input from these regions in addition to orexin neurons (Supplementary Fig. 3).

**Inhibition of the orexin$^{LH \to LC}$ pathway reduces cued fear expression.** To evaluate the function of the pathway identified by the cTRIO (Fig. 3), we examined whether optogenetic silencing of the orexin$^{LH \to LC}$ pathway during CS presentation affects freezing behavior. We bilaterally injected AAV$_{10}$-EF1α-FLEX (loxP)-vLWO-EGFP (vLWO, inhibitory Gi/o coupling vertebral cone opsin)[23] into the LH of Orexin-Cre mice, in which orexin neurons specifically express Cre recombinase[24]. We unilaterally implanted an optic fiber in the LC to test the effect of optogenetic inhibition of these axon terminals during cued fear test (Fig. 4a, b, g). We confirmed the expression of vLWO ($75.24 \pm 5.58\%$ orexin neurons expressed vLWO) and the position of optic fiber after each experiment, and only used the data when vLWO was specifically expressed in orexin neurons and optic fibers were adjacent to the LC (Fig. 4c). Following conditioning (Orexin-Cre-;vLWO, $n = 4$; Orexin-Cre+;vLWO, $n = 6$: two-way RM ANOVA with Sidak's post-hoc test, $F_{(1, 8)} = 0.3154$, $p = 0.5898$, Fig. 4d), optogenetic inhibition of orexinergic fibers in the LC during CS representation attenuated freezing, with the effect again more evident in the later phase of CS presentation (freezing over time: two-way RM ANOVA with Sidak's post-hoc test, $F_{(1, 8)} = 12.68$, $p = 0.0074$, Fig. 4e, left; average freezing: unpaired two-tailed Student's $t$ test, $t = 4.400$, $p = 0.0023$, Fig. 4e, right). Auditory CS or laser stimulation without prior conditioning had no effect on freezing behavior (one-way ANOVA, $F_{(7, 32)} = 1.049$, $p = 0.4179$, Fig. 4f). Furthermore, optogenetic silencing had no effect when the optic fibers were not correctly implanted above the LC (Supplementary Fig. 4). Fos induction in NA$^{LC}$ neurons by the CS was significantly decreased with the optogenetic inhibition (Orexin-Cre-; vLWO, $n = 3$; Orexin-Cre+;vLWO, $n = 5$: unpaired two-tailed Student's $t$ test, $t = 4.919$, $p = 0.0218$, Fig. 4h, i). Because orexin neurons are physiologically activated by fear-related cues (Supplementary Fig. 1), they are likely to lead to excitation of NA$^{LC}$ neurons. Our results suggest that inhibitory manipulation of this excitation impairs behavioral expression of fear.

**Upregulation of orexin signaling in the LC augments freezing.** Given that the orexin neurons are activated during behavioral fear expression (Supplementary Fig. 1), and orexin$^{LH \to LC}$ pathway plays a role in enhancing and sustaining freezing behavior, it might be also possible to induce or modulate fear-related behavior by artificially stimulating this circuit. To test this hypothesis, we injected AAV$_{10}$-EF1α-FLEX (loxP)-ChR2-EYFP (ChR2) into the LH of Orexin-Cre mice to express ChR2 specifically in orexin neurons ($74.49 \pm 5.35\%$ orexin cells expressed ChR2), and unilaterally implanted an optic fiber at the LC to activate axonal terminals of orexin neurons (Fig. 5a, b). We confirmed the expression of ChR2 after each experiment. We observed selective expression of ChR2 in orexin neurons as well as many ChR2-EYFP-positive fibers in the LC (Fig. 5c). Twenty-four hours after conditioning in "context A" in which both groups showed similar

**Fig. 5** Optogenetic excitation of orexin$^{LH \to LC}$-induced freezing behavior in Context A'. **a** Experimental procedure. **b** Schematic representation showing sites of AAV$_{10}$-FLEX-ChR2-EYFP delivery and optic fiber implantation in Orexin-Cre mice. **c** Focal expression of Cre-dependent ChR2 in orexin$^{LH}$ neurons (as shown with white arrow heads) and their axon terminals in the LC. Scale bars: upper panel, 100 μm; lower panel, 50 μm; right panel, 100 μm. **d, e** After cued fear conditioning, optogenetic stimulation of orexin $^{LH}$ to LC was applied in context A' for 150 s. **f** Control experiments indicating that laser stimulation did not show any effect on freezing without fear conditioning. **g** Optogenetic stimulation of orexin fibers induced robust Fos expression in NA$^{LC}$ neurons (white arrow heads). Scale bars: right panel, 100 μm. **h** Fos$^+$/TH+cells were markedly increased by optogenetic manipulation as compared to the control group (Orexin-Cre-;ChR2) and the WT (CS+) groups. **i, j** After cued fear conditioning, SB334867 administration 1 h before test session blocked optogenetically induced increase of freezing. **k** Laser stimulation itself did not show any effect on freezing without prior fear learning. *$p < 0.05$, **$p < 0.01$, ***$p < 0.001$. Values are presented as mean $\pm$ SEM

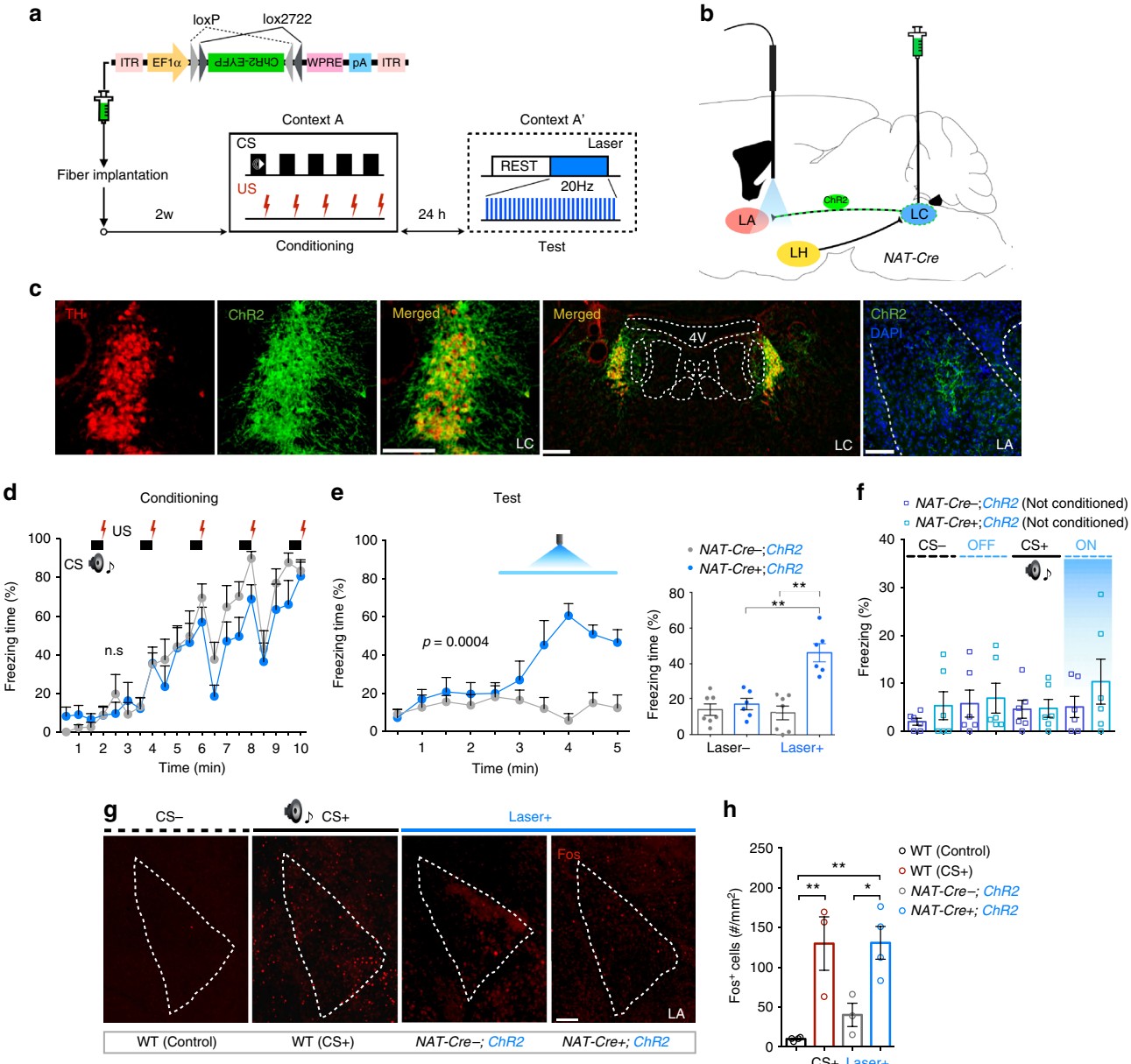

**Fig. 6** Optogenetic stimulation of NA$^{LC\rightarrow LA}$-induced freezing behavior in Context A′. **a** Experimental procedure. **b** Schematic representation showing sites of *AAV$_{10}$-FLEX-ChR2-EYFP* delivery and optic fiber implantation. **c** Focal expression of Cre-dependent ChR2 in NA$^{LC}$ neurons and their axonal terminals in the LA. Scale bars: left, 100 μm; center, 200 μm; right, 200 μm. **d**, **e** After cued fear conditioning, optogenetic stimulation of NA fibers in the LA for 150 s was applied in an altered context (A′). A significant increase in freezing time was induced by laser stimulation without applying auditory CS. **f** Control experiment shows that an increased freezing response induced by laser stimulation was not observed in mice without prior fear conditioning. **g**, **h** Optogenetic stimulation of NA$^{LC\rightarrow LA}$ fibers induced robust Fos expression in the LA, to the same degree as that in WT with CS. Scale bars: upper panel, 100 μm. *$p < 0.05$, **$p < 0.01$. Values are mean ± SEM

freezing level (*Orexin-Cre-;ChR2*, $n = 7$; *Orexin-Cre+;ChR2*, $n = 5$: two-way RM ANOVA with Sidak's post-hoc test, $F_{(1, 10)} = 0.0026$, $p = 0.9600$, Fig. 5d), we put these mice into a similar but distinct context (context A′), and optogenetically excited orexin fibers in the LC in the absence of the auditory CS. Context A′ normally did not evoke freezing behavior as shown in first 150-s of Fig. 5e. However, following the 150-s baseline phase, 150-s laser stimulation of orexin fibers in the LC evoked robust freezing behavior (freezing over time: two-way RM ANOVA with Sidak's post-hoc test, $F_{(1, 10)} = 23.38$, $p = 0.0007$, Fig. 5e, left, Movie Fig. 1; average freezing: unpaired two-tailed Student's *t* test, $t = 5.450$, $p = 0.0010$, Fig. 5e, right).

Freezing was not evoked in mice without prior conditioning (*Orexin-Cre-;ChR2* (not conditioned), $n = 7$; *Orexin-Cre+;ChR2* (not conditioned), $n = 5$: one-way ANOVA with Tukey's post-hoc test, $F_{(7, 40)} = 0.9152$, $p = 0.5049$, Fig. 5f). To assess the effectiveness of optogenetic stimulation, we confirmed an increase of Fos$^+$/TH$^+$ cells in NA$^{LC}$ neurons (WT (control), $n = 4$; WT (CS+), $n = 3$; *Orexin-Cre-;ChR2*, $n = 3$; *Orexin-Cre+;ChR2*, $n = 3$: one-way ANOVA with Tukey's post-hoc test, $F_{(3, 9)} = 14.82$, $p = 0.0008$, Fig. 5g, h). In addition, optogenetic stimulation did not affect behavior unless the optic fibers were precisely implanted above the LC region (Supplementary Fig. 5). Finally, we examined whether freezing responses induced by the orexin$^{LH\rightarrow LC}$

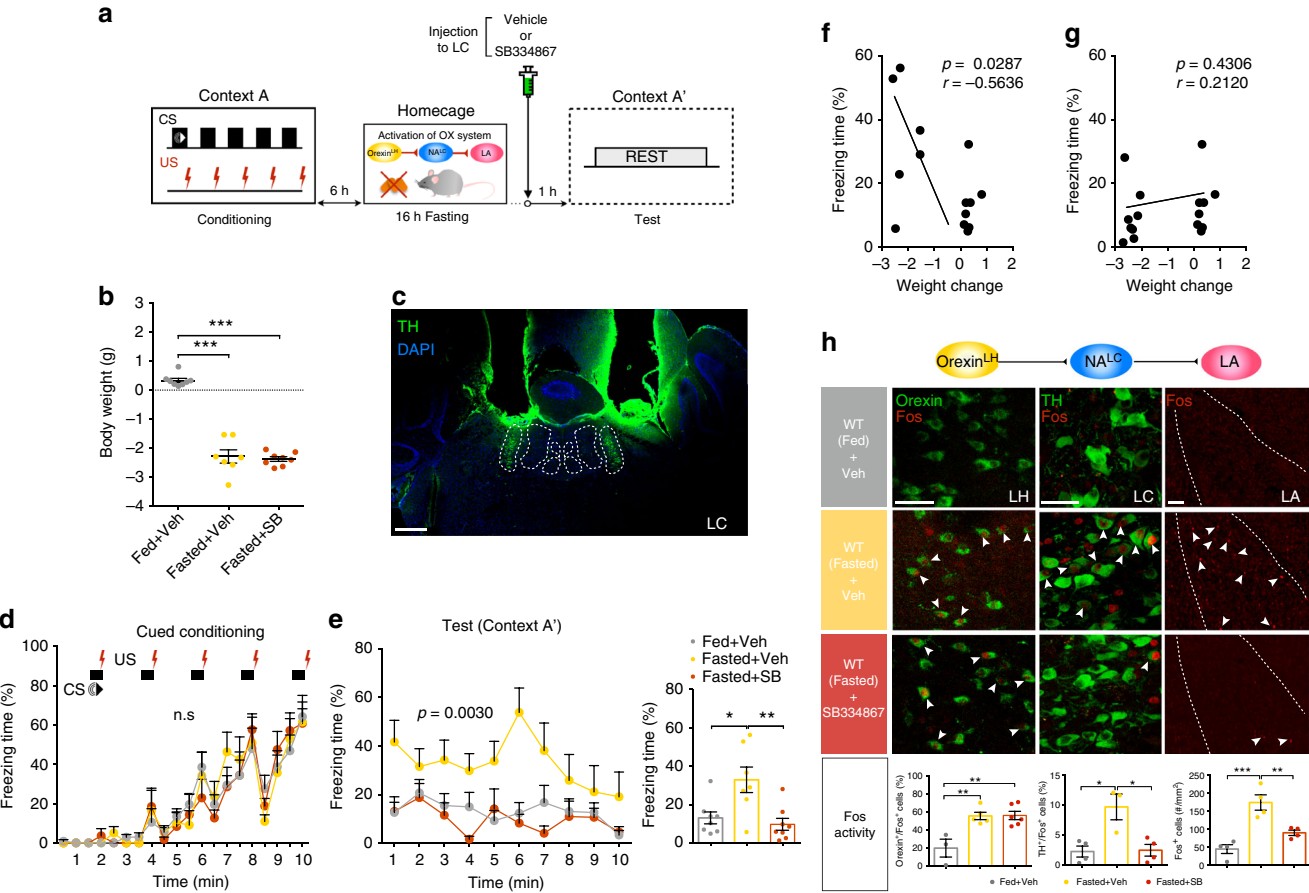

**Fig. 7** Physiological activation of orexin[LH] to NA[LC] pathway increased freezing behavior. **a** Experimental procedure. **b** Mouse's body weight was significantly decreased after fasting. **c** Representative image showing the bilateral placement of stainless cannula above the LC. **d** Gradual increase of freezing was observed during conditioning. **e** Six hours after cued fear conditioning following fasting for 16 h, mice showed increased freezing response in context A′ compared to the fed groups, and this response was attenuated by injection of SB334867 in the LC 1 h before the test session. **f** There were negative correlation between freezing time and weight change of Fed + Veh and Fasted + Veh groups. **g** There were no correlation between freezing time and weight change of Fed + Veh and Fasted + SB groups. **h** Orexin neurons, NA[LC] neurons, and LA neurons were highly activated by fasting condition, and focal administration of SB334867 into the LC inhibited activation of NA[LC] and downstream LA neurons. Images show the Fos+/Orexin+ cells in orexin neurons (left column, scale bar: 50 μm), Fos+/TH+ cells in NA[LC] neurons (center column, scale bar: 50 μm), Fos+ cells in the LA region (mm$^2$) (right column, scale bar: 100 μm) in each group (Fed + Veh, upper; Fasted + Veh, middle; Fasted + SB, bottom, Fig. 7h). Bottom figures showed the percentage of these cells in different regions regarding Fed + Veh, Fasted + Veh, Fasted + SB groups (Fig. 7h). *$p < 0.05$, **$p < 0.01$, ***$p < 0.001$. Values are mean ± SEM

stimulation require OX1R signaling. Twenty-four hours after fear conditioning (*Orexin-Cre+;ChR2* (vehicle), $n = 4$; *Orexin-Cre+; ChR2* (SB334867), $n = 5$: two-way RM ANOVA with Sidak's post-hoc test, $F_{(1, 7)} = 0.1756$, $p = 0.6877$, Fig. 5i), SB334867 or vehicle was administered i.p. to *Orexin-Cre* mice expressing ChR2 in orexin neurons. One hour after drug administration, we optogenetically stimulated orexin fibers in the LC. The photostimulation-induced freezing response was abolished in the SB334867-treated group (freezing over time: two-way RM ANOVA with Sidak's post-hoc test, $F_{(1, 7)} = 20.61$, $p = 0.0027$, Fig. 5j, left; average freezing: unpaired two-tailed Student's *t* test, $t = 3.972$, $p = 0.0144$, Fig. 5j, right), suggesting this response is mediated through OX1R. Also, photostimulation itself did not induce any behavioral change without prior cued fear conditioning (*Orexin-Cre−;ChR2* (not conditioned), $n = 4$; *Orexin-Cre+; ChR2* (not conditioned), $n = 5$: one-way ANOVA with Tukey's post-hoc test, $F_{(7, 28)} = 0.7448$, $p = 0.6366$, Fig. 5k), showing that activation of the orexin[LH→LC] signaling evokes fear expression only when the aversive US is already paired with a similar, but distinct context.

**Stimulation of NA signaling in the LA enhances freezing.** Because activation of orexin fibers in the LC evoked an enhanced fear response in a distinct but similar context (Fig. 5), and orexin neurons make monosynaptic contacts on NA[LC→LA] neurons (Fig. 3), NA[LC→LA] neurons might play an important role as a downstream effector in orexin neurons-evoked fear generalization. Thus, we next tested whether excitation of NA fibers in the LA also affect fear-related responses (Fig. 6a). We selectively restricted ChR2 expression in NA[LC] neurons by bilateral injection of *AAV₁₀-EF1α-FLEX (loxP)-ChR2-EYFP* into the LC of *NAT-Cre* mice (Fig. 6b) and unilaterally implanted an optical fiber in the LA. We confirmed focal expression of Cre-dependent ChR2 in NA[LC] neurons (83.40 ± 3.18%) and fibers in the LA (Fig. 6c). Twenty-four hours after fear conditioning in context A (*NAT-Cre−;ChR2*, $n = 7$; *NAT-Cre+;ChR2*, $n = 6$: two-way RM ANOVA with Sidak's post-hoc test, $F_{(1, 11)} = 0.9080$, $p = 0.3611$, Fig. 6d), we illuminated NA fibers in the LA in an altered context (A′) without applying the auditory CS. After a 150-s rest phase, 150-s laser stimulation of NA fibers in the LA markedly increased freezing (freezing over time: two-way RM ANOVA with Sidak's

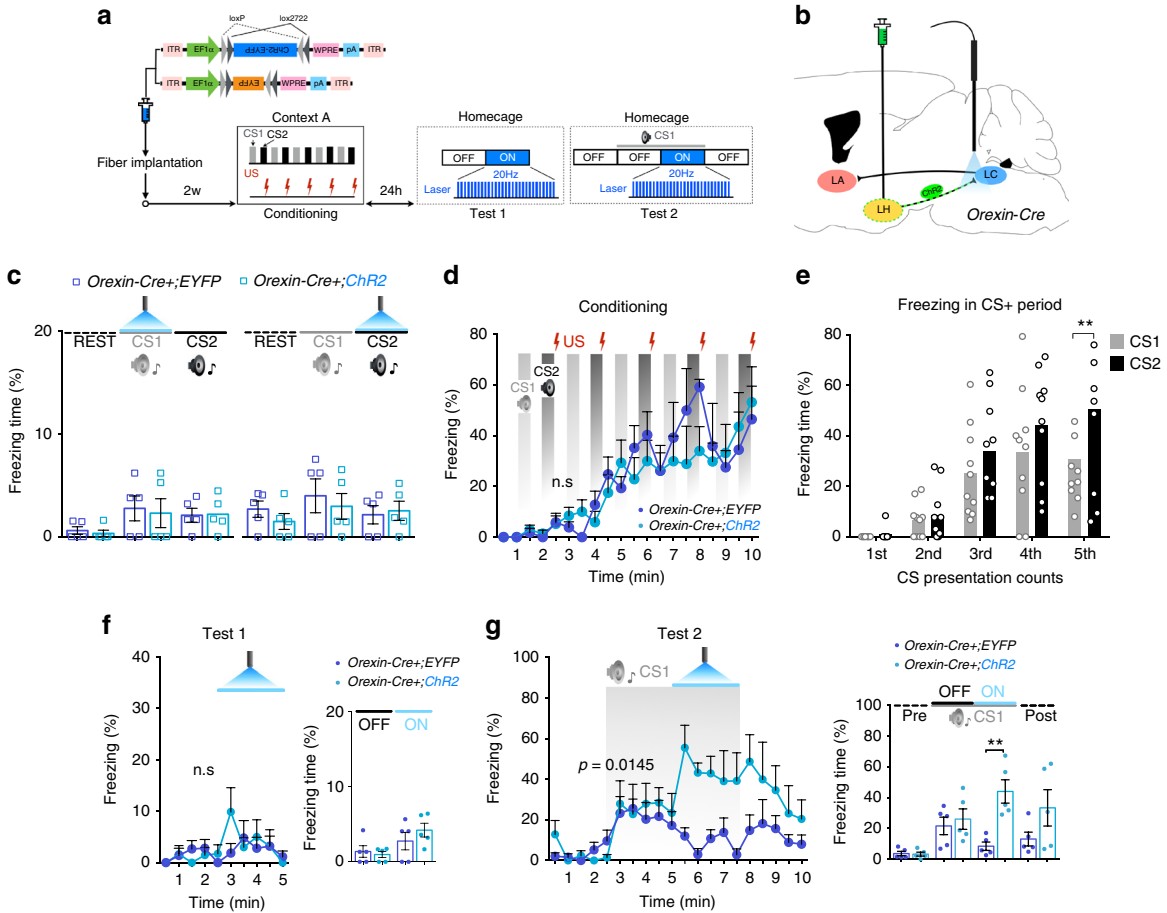

**Fig. 8** Optogenetic simulation of orexin$^{LH \to LC}$ pathway potentiates CS-triggered fear expression and generalization. **a** Experimental procedure. **b** Schematic representation showing sites of *AAV-EF1α-FLEX (loxP)-ChR2-EYFP* and *AAV-EF1α-FLEX (loxP)-EYFP* delivery and optic fiber implantation. **c** Before conditioning, CS1(white noise) or CS2 (tone), even coupled with optogenetic stimulation of orexin$^{LH \to LC}$ fibers did not affect freezing response. **d, e** In cued fear conditioning session, although freezing time was gradually increased, different type of CS (CS1, CS2 paired with US) showed differential effects on freezing. **f** Twenty-four hours after fear conditioning, optogenetic stimulation of orexin$^{LH \to LC}$ pathway in homecage did not show any freezing response in contrast with that observed in context A′ (Fig. 5e). **g** We performed optogenetic stimulation with CS1 in homecage. During the presentation of CS1, optogenetic stimulation of orexin$^{LH \to LC}$ pathway strongly potentiated freezing behavior induced by CS1 as compared to control group. *$p < 0.05$, **$p < 0.01$. Values are mean ± SEM

post-hoc test, $F_{(1, 11)} = 14.30$, $p = 0.0030$, Fig. 6e, left, Movie Figs. 2 and 3; average freezing: unpaired two-tailed Student's $t$ test, $t = 5.279$, $p = 0.0004$, Fig. 6e, right). Although we did not monitor firing rate of NA$^{LC}$ neurons in fearful condition, Takeuchi et al reported that these neurons showed burst firing at ranging from 15.3 to 28.2 Hz when the mice were put into the novel cage, although basal firing rate is 0.23–6.03 Hz[25]. Based on this observation, we here used pulses of 473 nm illumination at 20 Hz (pulse width = 35 ms, interval = 15 ms, 10 mW) for the photostimulation. Again, we did not see a photostimulation-induced freezing response in context A′ without prior fear conditioning in context A (*NAT-Cre-;ChR2* (not conditioned), $n = 6$; *NAT-Cre+;ChR2* (not conditioned), $n = 6$: one-way ANOVA with Tukey's post-hoc test, $F_{(7, 40)} = 0.7440$, $p = 0.6364$, Fig. 6f). After optical stimulation, we validated Fos-positive cells in the LA. Fos-positive cells were increased by optogenetic manipulation (WT (control), $n = 4$; WT (CS+), $n = 3$; *NAT-Cre-;ChR2*, $n = 3$; *NAT-Cre+;ChR2*, $n = 4$: one-way ANOVA with Sidak's post-hoc test, $F_{(3, 10)} = 10.66$, $p = 0.0019$, Fig. 6g, h), suggesting that the noradrenergic influence on LA neurons is excitatory. When optic fibers were not precisely implanted at the LA, photostimulation did not evoke any effect in the conditioning and test sessions (Supplementary Fig. 6).

The maximum freezing time observed with optogenetic activation was comparable to the CS-evoked freezing time in WT mice (Supplementary Fig. 7a), although the responses were usually delayed as compared with those induced by auditory cues (Supplementary Fig. 7b), presumably due to the slow neurotransmission governed by the peptidergic and monoaminergic systems. These observations suggest that with the optogenetic stimulation of orexin fibers in the LC or NA fibers in the LA, mice responded to trace elements of the contexts, which normally are not recognized by mice as danger, facilitating the expression of learned fear responses against circumstances that differ from those during acquisition of the fear memory[26]. This is akin to the response observed in PTSD patients, raising the possibility that over-activation of NA$^{LC}$ neurons could underlie the pathological behavioral response, such as fear generalization, seen in PTSD or panic disorder.

**Physiological increase of orexinergic tone modulates freezing.** Our various manipulations of the circuits involving orexin$^{LH \to LC}$ and NA$^{LC \to LA}$ neurons established that this pathway is important for the proper expression of freezing behavior. However, it is not clear whether a physiological activation of this

neural circuit could actually influence animals' behavior in a real-world situation. To explore the effects of physiological activation of orexin neurons on fear-related behavior, we used fasting, a manipulation which is well known to activate orexin neurons[27–30]. Six hour after cued fear conditioning in the context A, mice were deprived of food for 16 h and then subjected to a test in context A′ in the absence of the CS (Fig. 7a). Fasting significantly changed the mouse's body weight (Fed + Veh, $n = 8$; Fasted + Veh, $n = 7$; Fasted + SB, $n = 8$: one-way ANOVA with Tukey's post-hoc test, $F_{(2, 20)} = 129.1$, $p < 0.0001$, Fig. 7b). We implanted stainless cannulae bilaterally above the LC for acute SB334867 injection (Fig. 7c). After cued fear conditioning (two-way RM ANOVA with Tukey's post-hoc test, $F_{(2, 20)} = 0.02$, $p = 0.9802$, Fig. 7d) and fasting, mice showed increased freezing response even in a distinct but similar context (context A′). We also found that the fasting-induced freezing was attenuated with an OX1R antagonist, SB334867 focally injected in the LC (freezing over time: two-way RM ANOVA with Tukey's post-hoc test, $F_{(2, 20)} = 7.850$, $p = 0.0030$, Fig. 7e, left; average freezing: one-way ANOVA with Tukey's post-hoc test, $F_{(2, 20)} = 7.852$, $p = 0.0030$, Fig. 7e, right), suggesting that OX1R in the LC is involved in enhanced fear expression induced by fasting. The negative correlation between weight change and freezing response disappeared in fasted mice when administered SB334867 in the LC ($n = 15$, $r = -0.5636$, $p < 0.05$; $n = 16$, $r = 0.2120$, $p = 0.4306$, Fig. 7f, g). Neuronal activity evaluated with Fos in each component of the circuit was elevated following fasting, suggesting this circuit is engaged during freezing in context A′. The activation of NA[LC] neurons and LA neurons by fasting were inhibited with administration of SB3348687 in the LC without any change in the activity of orexin neurons, suggesting that orexin[LH→LC] to NA[LC→LA] circuit is mobilized by freezing (orexin; Fed + Veh, $n = 3$; Fasted + Veh, $n = 5$; Fasted + SB, $n = 6$: one-way ANOVA with Sidak's post-hoc test, $F_{(2, 11)} = 10.69$, $p = 0.0026$; NA[LC]; Fed + Veh, $n = 4$; Fasted + Veh, $n = 3$; Fasted + SB, $n = 4$: one-way ANOVA with Tukey's post-hoc test, $F_{(2, 8)} = 9.596$, $p = 0.0075$; LA; Fed + Veh $n = 4$; Fasted + Veh, $n = 4$; Fasted + SB, $n = 4$: one-way ANOVA with Tukey's post-hoc test, $F_{(2, 9)} = 19.61$, $p = 0.0005$, Fig. 7h). Similar result was obtained in the experiment with i.p. injection of SB334867 (Supplementary Fig. 8).

**Activation of orexin signaling in the LC augments cued fear response.** Optogenetic stimulation of orexin[LH→LC] or NA[LC→LA] pathways resulted in a robust increase of freezing in context A′ without CS (Figs. 5 and 6). We postulated that this freezing was evoked by similar contextual elements between context A′ and context A. We further examined whether explicit cues that are normally are not recognized as danger, but contains similar elements also induce freezing when paired with excitation of the orexin[LH→LC] pathway. To achieve this, we presented mice with two different auditory CSs, one never paired with shock and one always paired with shock (CS1, white noise and CS2, tone) (Fig. 8a, b). We injected $AAV_{10}$-EF1α-FLEX (loxP)-ChR2-EYFP or $AAV_{10}$-EF1α-FLEX (loxP)-EYFP into the LH of Orexin-Cre mice to specifically express ChR2 or EYFP in orexin neurons. Prior to conditioning, neither CS1 nor CS2 triggered freezing behavior, even when paired with optogenetic simulation of the orexin[LH→LC] pathway (Orexin-Cre+;ChR2-EYFP, $n = 5$; Orexin-Cre+; ChR2, $n = 5$: one-way ANOVA with Tukey's post-hoc test $F_{(5, 24)} = 1.260$, $p = 0.3131$, $F_{(5, 24)} = 0.5857$, $p = 0.7107$, Fig. 8c). We did not observe significant differences in freezing between ChR2-injected and EYFP-injected mice during training (two-way RM ANOVA with Sidak's post-hoc test $F_{(1, 8)} = 0.5120$, $p = 0.4946$, Fig. 8d) with both groups showing significantly more freezing to

CS2 compared to CS1 (CS1, $n = 10$; CS2, $n = 10$: two-way RM ANOVA with Sidak's post-hoc test $F_{(1, 9)} = 6.593$, $p = 0.0303$ Fig. 8e). After conditioning, we performed testing with optogenetic manipulation in the homecage to exclude any influence by contextual elements. Notably, optogenetic activation of the orexin[LH→LC] pathway in the absence of any CS did not evoke freezing (freezing over time: two-way RM ANOVA with Sidak's post-hoc test $F_{(1, 8)} = 0.2447$, $p = 0.6341$, Fig. 8f, left; average freezing: one-way ANOVA with Tukey's post-hoc test $F_{(3, 16)} = 2.949$, $p = 0.0643$ Fig. 8f, right). CS1 presentation alone induced small increase in freezing in both groups. However, laser stimulation during CS1 application led to a large increase in freezing in mice expressing ChR2 in orexin neurons (two-way RM ANOVA with Sidak's post-hoc test $F_{(1, 8)} = 9.648$, $p = 0.0145$, Fig. 8g, left). A comparison of average freezing times revealed that ChR2-expressing mice compared to EYFP controls only showed elevated freezing when laser stimulation was applied during CS1 presentation (one-way ANOVA with Tukey's post-hoc test $F_{(7, 32)} = 5.682$, $p = 0.0003$, Fig. 8g, right). These results suggest that the orexin[LH→LC] pathway modulates the behavioral expression of fear against an auditory cue.

## Discussion

Our present study establishes that orexin[LH→LC] and NA[LC→LA] neurons form a circuit, which plays a critical role in the regulation of freezing behavior. Our data integrate previous findings to provide a deeper understanding of the roles of orexin neurons in emotional responses and suggest that these neurons not only increase the sympathetic nervous outflow[31] and arousal, but also play a crucial role in ensuring the expression of an appropriate behavioral response. We demonstrated that orexin neurons are strongly activated by fearful situations[32] (Supplementary Fig. 1), which is likely to lead to activation of downstream pathways, including NA[LC→LA] neurons. Genetic or pharmacological blockade of OX1R, a receptor abundantly expressed in NA[LC] neurons, resulted in a significant reduction of CS-mediated freezing. Although we could not compare result from two separated experiments, the effect of the antagonist seems to be larger as compared with local deletion of OX1R in NA[LC] neurons (Fig. 1g), suggesting that another pathway (s) involving OX1R signaling might play an additional role.

A previous study reported that acute blockade of OX1R did not affect fear memory retrieval during random presentations of a 30-s CS[17]. Our results are consistent with this, as neither inhibitory antagonist administration, nor inhibitory manipulations with pharmacogenetics and optogenetics affected freezing time during the initial 30-s of CS presentation. Rather, these manipulations resulted in a reduction of freezing response in the later phase of CS presentation (Figs. 1, 2, and 4), suggesting the function of orexin system in sustaining the behavioral expression of fear rather than the expression of fear-related behavior itself. Furthermore, the function of orexin in the regulation of fear-related behavior is consistent with the previous observation that human narcolepsy patients, who lack orexin signaling, have an impaired fear response and show reduced amygdala activity when exposed to aversive conditioned stimuli[33,34].

In this study, we confirmed that orexin neurons make direct synaptic contact with NA[LC→LA] neurons (Fig. 3), suggesting the existence of orexin[LH→LC] to NA[LC→LA] pathway. SADΔG is likely to be effectively transported from post-synaptic neurons to pre-synaptic neurons only when the synapses between these two neurons are tight. Although peptidergic neurons can form tight synapses, the majority are thought to be loose synapses, thus we speculate that the number of labeled orexin neurons in our cTRIO experiment (Fig. 3) might be an underestimate of the

actual connectivity. Orexin neurons have broad projections to targets employing diverse post-synaptic receptors and signaling complexes, thus they are also well suited to mediate other various aspects of the fear responses, including freezing, cardiovascular function, and stress response in response to emotionally salient stimuli[3,35,36]. Among them, we found that circuit composed of orexin[LH→LC] and NA[LC→LA] neurons plays an important role in regulation of fear-related behavior. Activation of this circuit increased freezing behavior even in a context distinct from the training context (Figs. 3 and 4). With the optogenetic activation, mice recognize elements common to both contexts as signal that is associated with dangerous outcomes. This signal might be normally attenuated by top–down inhibitory signals from regions such as the prefrontal cortex[37], but becomes evident when the orexin system is mobilized. Alternatively, NA signaling in the LA might increase attention to trace elements that suggest danger. After fear conditioning in a given context (such as context A), mice tend to exhibit slightly increased freezing (less than 20%) in an altered but similar context (A′), suggesting that the animals recognize some elements of this context as cues that predict aversive outcome (Supplementary Fig. 9). Increased activation of the orexin[LH→LC] or NA[LC→LA] circuits enhanced the fear response in context A′, revealing the ability of this circuit to open the "gate of fear." In other words, the increase of the activity in these circuits might shift an anxiety state to one of actual fear. Consistent with this, optogenetic simulation of orexin[LH→LC] did not increase freezing in the homecage following fear conditioning (Fig. 8f). This supports the hypothesis that NA[LC→LA] circuit gates fear against trace elements, which are normally ignored as indicators of danger and orexin regulates this activity. Furthermore, we observed similar results when we examined the impact of excitatory manipulation of orexin[LH→LC] circuit during the presentation of an auditory cue that was not explicitly predictive of the US (Fig. 8g).

The optogenetic manipulations on orexin[LH→LC] or NA[LC→LA] circuits resulted in fear generalization, a process of emergence of fear against a stimulus or context to related cues or similar situations previously paired with aversive experiences. These observations raise the possibility that activation of orexin neurons may underlie the internal condition that evokes fear generalization. From another perspective, stressful conditions, such as fasting, could result in an orexin-dependent shift to a "vigilance" state and alter the selection of a behavioral response to avoid and handle environmental threats in real-world situation.

Our study also suggests that the inhibition of OX1R may provide a promising avenue for treating psychiatric conditions that are characterized by exaggerated and/or inappropriate fear-related responses and anxiety triggered by external cues, such as panic disorder and PTSD.

## Methods

**Animals**. All experimental procedures involving mice were approved by the Animal Experiment and Use Committee of University or Kanazawa University (AP-111947) and Tsukuba University (16–397), and were thus in accordance with NIH guidelines. We used BAC-transgenic *NAT-Cre* mice[21], and *Orexin-Cre* mice[24] for Cre-dependent pharmacogenetic and optogenetic manipulation. We confirmed highly specific expression of Cre recombinase in NA[LC] neurons by mating *NAT-Cre* mice with *Rosa26Sor[tm1sor]* reporter mice and did not detect any ectopic expression in other regions (Supplementary Fig. 2b). We generated *OX1R[f/f];NAT-Cre* mice (Supplementary Fig. 2), in which OX1R is selectively deleted in NA[LC] neurons, for cued fear conditioning experiments. Mice were maintained under a strict 12-h light–dark cycle in a temperature- and humidity-controlled room and fed ad libitum.

**Viral preparation**. We used recombinant AAV vectors with the FLEX switch[38] system to specifically express ChR2 or vLWO, hM4Di (DREADD) in Cre-expressing neurons. We used long-wavelength vertebral cone opsin (vLWO) for optogenetic inhibition. vLWO activates the Gi/o pathway in neurons to activate

Kir3 channels[23]. Human red opsin (vLWO) cDNA clone was excised from a plasmid *pAAV-CMV-FLEX (loxP)-vLWO*. The excised vLWO was cloned into *pAAV-EF1α-FLEX (loxP)-hChR2 (H134R)-EYFP-WPRE-pA*. We used *pAAV-EF1α-FLEX (loxP)-hM4Di(Gi)-mCherry* (provided by Dr Bryan Roth), *pAAV-EF1α-FLEX (loxP)-hChR2(H134R)-EYFP-WPRE-HGHpA* (provided by Dr Karl Deisseroth), *pAAV-EF1α-FLEX (loxP)-EYFP-WPRE-HGHpA* (Addgene plasmid # 20296). *pAAV-CAG-FLEX (Frt)-TVA-mCherry* and *pAAV-CAG-FLEX (Frt)-RG* (rabies glycoprotein) were provided by Dr Miyamichi (Tokyo University). All viral plasmids were packed with *pHelper* (Stratagene) and *pAAV2-rh10* (provided by Penn Vector Core) with a triple transfection, helper-free method[39]. We also purchased CAV2-FLEX (loxP)-Flp virus from Biocampus Montpellier. *pcDNA-SADB19L* (#32632), *pcDNA-SADB19G* (#32633), *pcDNA-SADB19N* (#32630), *pcDNA-SADB19P* (#32631), and *pSADdeltaG-GFP-F2* (#32635) were obtained from Addgene. We made SADdΔG-GFP(EnvA) by transfecting these plasmids in B7GG cells, followed by pseudotyping in BHK-RGCD-EnvA cells, and ultracentrifugation[40]. The titers of virus vectors were: *AAV₁₀-EF1α-FLEX (loxP)-ChR2-EYFP*; $3.7 \times 10^{13}$, *AAV₁₀-EF1α-FLEX (loxP)-hM4Di-mCherry*; $1.90 \times 10^{12}$, *AAV₁₀-EF1α-FLEX (loxP)-vLWO- EGFP*; $7.04 \times 10^{12}$, *AAV₁₀-EF1α-FLEX (loxP)-EYFP*; $5.82 \times 10^{13}$ (genomic copies/ml), *CAV2-FLEX (loxP)-Flp* (purchased from Biocampus Montpellier), *AAV₁₀-CAG-FLEX (Frt)-TVA-mCherry*; $4 \times 10^{13}$, *AAV₁₀-CAG-FLEX (Frt)-RG*; $3.9 \times 10^{13}$ (genome copies/ml), and *SADΔG-EGFP (EnvA)*; $4.2 \times 10^{8}$ infectious units/ml.

**Drugs**. The orexin receptor-1 antagonist, SB334867 (Tocris) (5 mg/kg), was dissolved in 1% dimethyl sulfoxide in distilled water. SB334867 and vehicle were injected i.p. 1 h before the cued fear test session of fear conditioning (Fig. 1e–g). Same drug and procedure was used for optogenetic experiment to activate orexin[LH→LC] neurons (Fig. 5i–k), and other experiment with combination of fasting and fear conditioning (Supplementary Fig. 8). For microinjection experiment, we used SB334867 in physiological saline (10 nmol/0.5 µl) or saline (Veh) 30 min before test session (Fig. 7). All these doses were based on previous study[18]. For hM4Di (DREADD) experiments, clozapine-N-oxide (CNO) (5 mg/kg) or saline was injected i.p. 40 min before the test session (Fig. 2a).

**Surgery**. For the inhibitory DREADD experiment, *NAT-Cre* mice (12 weeks old) were anesthetized with sodium pentobarbital (0.5 mg/kg, i.p.) and positioned in a stereotaxic frame (David Kopf Instruments). *AAV₁₀-EF1α-FLEX (loxP)-hM4Di-mCherry* was injected into the LC (AP, −5.4 mm; ML, ± 0.9 mm; DV, −3.65 mm; 0.6 µl in each site) over a 10-min period. For optogenetic manipulation of orexin neurons, *Orexin-Cre* mice were injected with *AAV₁₀-EF1α-FLEX (loxP)-vLWO-EGFP* or *AAV₁₀-EF1α-FLEX (loxP)-ChR2-EYFP* bilaterally into the LH (AP, −1.4 mm, −1.8 mm; ML, ±0.9 mm; DV, −5.5, −5.7; 0.25 µl in each site). Then, an optical fiber was implanted above the LC (AP, −5.4 mm; ML, +0.9 mm; DV, −3.3 mm). For optogenetic activation of NA[LC] neurons, *NAT-Cre* mice were injected with *AAV₁₀-EF1α-FLEX (loxP)-ChR2-EYFP* bilaterally into the LC (same sites as DREADD experiment; 0.8 µl in each site). Then, an optical fiber was implanted at the LA (AP, −1.8 mm; ML, +3.25 mm; DV, −3.75 mm). After a 14-day recovery period in individual cages after injection, mice were subjected to behavioral experiments, and then killed and brain samples were collected for immunohistochemical examination. Behavioral data were only included if these viruses were targeted specifically and the fiber optic implants were precisely placed. For microinjection of SB334867 directly into the LC, we bilaterally implanted stainless cannula just above the LC (AP, −5,4 mm; ML, ±0.9 mm; DV, −3.2).

**Behavioral experiment**. All experiments were performed during the light phase (13:00–16:00) using male 12 to 14-week-old mice. Prior to the experiments, these mice were isolated for 2 weeks. The experimenters were blind to the genotypes until all data had been gathered and analyzed. We performed same cued fear conditioning paradigms with modified retrieval test using longer CS presentation (150 s) than the previous protocols[8]. We used this condition, because we found this robustly excited NA[LC] neurons (Fig. 5). For the cued fear conditioning test, mice were placed in a conditioning chamber (Context A; 15 × 12 × 13 cm) for 90 s before giving a conditioned stimulus (CS), a 2900 Hz, 80 dB tone, which lasted 30 s, immediately followed by the presentation of an unconditioned stimulus (US), a mild foot shock of 0.3 mA for 2 s, on the training day. We also used 0.4 mA for 2 s in fasting experiment (Fig. 7 and Supplementary Fig. 8). Five consecutive sessions of training were performed. After an additional stay for 90 s in the chamber, the mouse was returned to its homecage. Test sessions were performed 24 h after the training. The test session was conducted in a chamber (Context A′; 13 × 20 × 30 cm) with a different context (a different cage surrounded by a monochrome striped screen), and freezing behavior was scored during the test session. A memory test was performed with a 150-s rest period following the presentation of 150 s auditory CS. For the optogenetic inhibition experiments with vLWO in orexin[LH] neurons, 3000 pulses of 473 nm illumination at 20 Hz (pulse width = 35 ms, interval = 15 ms, 10 mW) were presented during a 150-s period. In the optogenetic experiments with ChR2, light pulses with different strength (15 mW) were presented in the same manner for 150 s following a 150-s period without shining laser. Frequency of the stimulation was based on the previous study

focusing on the firing rate of LC neurons (15.3–28.2 Hz) when the mice were put into the novel context[25].

For experiment shown in Fig. 8, we conducted optogenetic stimulation of orexin neurons with two different types of CS. We presented 30 s CS1 (white noise, 70 dB) 30 s before giving a CS2 (2900 Hz, 80 dB), which lasted 30 s, that was immediately followed by an unconditioned stimulus (US, 0.3 mA). We repeated the sequence five times. Twenty-four hours after the fear conditioning, we performed 150 s optogenetic stimulation of orexin$^{LH \to LC}$ as described above, after 150 s of REST period in a homecage for recording the baseline (Test 1). Additionally, we performed 300 s CS1 presentation following 150 s REST, and illuminated the same pathway for 150 s in the latter half of CS1, then, we also observed the freezing behavior for 150 s after these manipulations (Test 2).

Freezing behavior was recorded with a charge-coupled device (CCD) video camera and analyzed with an auto tracking system, compACT VAS ver 3.0x (Muromachi Kikai, Tokyo). Data are presented as mean ± SEM.

**Immunohistochemistry**. Mice were anesthetized with sodium pentobarbital and then fixed by intracardiac perfusion with 4% paraformaldehyde. Then, the brain was post-fixed for 24 h in the same fixative and cryoprotected by immersion in 30% sucrose for 2 days. Brain sections of 30 μm thickness were cut with a cryostat. Sections were washed and blocked with 0.1 M phosphate-buffered saline (PBS) containing 0.25% Triton X-100 plus 3% bovine serum albumin (BSA). Then, slices were incubated with the designated primary antibodies in PBS overnight at 4 °C. Antibodies used in this study were guinea pig antibody against orexin B antibody (1:500), rabbit polyclonal antibody against Fos (1:5000, Ab-5, Millipore), mouse polyclonal antibody against tyrosine hydroxylase (1:2000, F-11, Santa Cruz), rat antibody against green fluorescent protein (GFP) (1:1000, Nakarai), goat polyclonal antibody against mCherry (1:500, SICGEN), and rabbit anti-beta galactosidase (Cappel 55796). Then, slices were washed with PBS three times, followed by incubation with designated secondary antibodies in PBS for 2.5 h. Secondary antibodies used in this study were Alexa 594-conjugated donkey anti-rabbit IgG, Alexa 594-conjugated donkey anti-mouse IgG, Alexa 594-conjugated donkey anti-guinea pig IgG, Alexa 488-conjugated donkey anti-mouse IgG, Alexa 488-conjugated donkey anti-rabbit IgG, Alexa 488-conjugated donkey anti-rat IgG, Alexa 488-conjugated donkey anti-guinea pig IgG, and Alexa 647-conjugated donkey anti-mouse IgG (1:1000; Molecular Probes, Eugene, OR). Slices were washed three times in PBS, mounted on subbed slides, air dried, and coverslipped using FluorSave Reagent (Calbiochem). For DAPI staining, slides were covered with DAPI-containing mounting reagent (Vector Laboratories). All images were taken on Olympus FV10i confocal microscope. The number of Fos$^+$/TH$^+$ cells in NA$^{LC}$ neurons were counted in every 30 μm coronal section throughout the LC. We also evaluated the activity of orexin neurons by counting Fos$^+$/Orexin$^+$ cells in the LH. Fos$^+$ cells in the LA were counted (cells per area) in coronal sections from −1.7 to −2.06 from the bregma using ImageJ. Cells were counted on both sides of these brain regions.

**In situ hybridization**. We performed in situ hybridization to detect OX1R mRNA in NA$^{LC}$ neurons to evaluate the deletion of OX1R mRNA expression in NA$^{LC}$ neurons of OX1R$^{f/f}$; NAT-Cre mice. Following the protocol described in previous study[41], we stained OX1R with in situ hybridization combined with TH immunohistochemistry. First, 30 μm floating brain sections were treated with 4% paraformaldehyde and 0.3% Triton-X. After rinsing with PBS, the sections were incubated in 1% NaBH$_4$ containing PBS for 30 min followed by 0.75% glycine in PBS (15 min, two times) and treated with 4% PFA. After rinsing with PBS, the sections were treated 0.5% acetic anhydrate for 15 min. After rinsing with PBS, we incubated the sections in hybridization buffer at 60 °C for 1.5 h followed by incubation with digoxigen-conjugated anti-sense OX1R RNA probes for 16 h at 60 °C. Then, sections were washed in 2× standard citrate saline (SSC) with 50% formamide/0.1% N-lauroylsarcosine reagent at 50 °C (20 min, two times) and incubated with RNase A treatment. After rinsing with 2× SSC with 0.1% N-lauroylsarcosine, 0.2× SSC with 0.1% N-lauroylsarcosine, and Tris-buffer saline (TBS, pH 7.5), the sections were treated with TBS (pH 7.5) containing 1% blocking reagent for 1.5 h. Then, the sections were incubated with alkaline phosphatase-conjugated anti-digoxigenin (1:2000, Roche) and TH (1:2000, F-11, Santa Cruz) antibodies O/N at 4 °C. After rinsing TBS (pH 7.5)/0.1% Tween 20 followed by incubation of TBS (pH 7.5)/1% blocking reagent, the sections were incubated with Alexa 488-conjugated donkey anti-mouse IgG (1:1000; Molecular Probes, Eugene, OR) for 1 h. After rinsing with TBS (pH 7.5)/0.1% Tween 20 followed by TBS (pH 8.0) with 50 mM Mg solution, the sections were treated with Fast Red solution enhanced by HNPP (HNPP Fluorescent Detection Set, Roche). The sections were then washed with TBS (pH 7.5) and mounted on slide glasses. Images were acquired with a Leica SP8 confocal microscope.

**cTRIO**. To identify the orexin$^{LH \to LC}$ to NA$^{LC \to LA}$ circuit, we used the trans-synaptic tracing (cTRIO) with combination of Cre- and Flp-dependent virus expression for rabies virus (SAD∆G-EGFP (EnvA))[19]. We used NAT-Cre mice for retrograde expression of Cre-dependent Flp in the LC region. Flp lead to

expression of Flp-dependent TVA and RG, which were necessary for expression and trans-synaptic transfer of SAD∆G-EGFP (EnvA) in upstream neurons. NAT-Cre mice were anesthetized with sodium pentobarbital (0.5 mg/kg, i.p.) and injected with 0.4 μl of CAV2-FLEX(loxP)-Flp into ipsi-lateral LA using coordinates described above. A 1: two mixture (0.6 μl) of AAV$_{10}$-CAG-FLEX(Frt)-TVA-mCherry and AAV$_{10}$-CAG-FLEX (Frt)-RG into the left LC. Two weeks later, 0.6 μl of SAD∆G-EGFP (EnvA) was injected into the LC using the procedure described above. Mice were kept in homecage for 5 days before killing. This manipulation labels NA$^{LC \to LA}$ cells (starter cells), and visualizes the upstream neuronal populations that make direct synaptic input to starter cells. We detected DAPI$^+$/Orexin$^+$/GFP$^+$ cells in the LH and starter cells with DAPI$^+$/mCherry$^+$/TH$^+$ cells in the LC. All images were taken on Olympus FV10i confocal microscope.

**Fasting**. Six hours after cued fear conditioning, mice were fasted for 16 h before the test session in context A′. During fasting, food was removed but water was available ad libitum. Control mice could access to food and water ad libitum. Weight change of mice was measured before and after fasting procedure.

**Statistical analysis**. All data were expressed as mean ± SEM. We used one-way or two-way analysis of variance (ANOVA) followed by Sidak's or Tukey's post-hoc tests for serial freezing responses. Fos-positive cell count, or Student's t test for average freezing responses and Fos-positive cell count. Pearson correlation coefficient was used for analyzing the strength of relationship between freezing vs. weight change, Orexin$^+$/Fos$^+$ cells vs. weight change, freezing vs. Orexin$^+$/Fos$^+$ cells. All statistics were performed using Graph Pad Prism 6.0b. Differences were considered significant at *$p < 0.05$, **$p < 0.01$, ***$p < 0.001$.

**Data availability**. The authors declare that all data supporting the findings of this study are available within the article and its supplementary information file.

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

## Acknowledgements

The authors thank Drs K. Miyamichi, N. Uchida, and H. Osakada for providing the cells and plasmids for viral tracing and valuable discussion, Mr Arthur J.Y. Huang and the RIKEN BSI RRC for *NAT-Cre* mouse generation and Dr Maya Yamazaki and Ms Rie Natsume for *OX1R^{f/f}* mouse generation. This study was supported by a Grant-in-Aid for JSPS Fellows (S.S), a KAKENHI Grant-in-Aid for Scientific Research on Innovative Areas, "Adaptive circuit shift" (JP15H01425) (T.S.), Scientific Research (B) (JP 15H03122) (T.S.), Exploratory Research (T.S.) (JP 15K12768), CREST, JST, "Opto-Bio" (T.S.), SENSHIN Medical Research Foundation (T.S.), the Merck Investigator Studies Program (#54843) (T.S.), Novartis Foundation (T.M.), RIKEN BSI (T.J.M.), Deutsche Forschungsgemeinschaft Grants He2471/21-1, He2471/18-1 Priority Program (SPP1926), SFB874 (B10) and SFB1200 (A07) (S.H.), and a KAKENHI Grant-in-Aid for Scientific Research on Innovative Areas "Willdynamics" (16H06401) (T.S.).

## Author contributions

S.S. designed the experiments, collected and analyzed all data, and wrote the manuscript. T.S. designed the experiments, prepared virus vectors, and wrote the manuscript. T.M.T. performed some pharmacological experiments and analyzed data (Fig. 1g). T.J.M. provided the *NAT-Cre* mouse line and their behavioral data, histological figure, and contributed to editing the manuscript. T.M. contributed to generate *AAV_{10}-FLEX-vLWO-EGFP*. S.H. provided cDNA for vLWO. M.A. and K.S. contributed to generate *OX1R^{f/f}* mice.

## Additional information

**Competing interests:** The authors declare no competing financial interests.

