## [Peer Review File · Nature Communications]

Reviewers' comments:

Reviewer #1 (Remarks to the Author):

This manuscript reports connectivity between LH orexin neurons, the LC as a major effector and the LA as an output of LC stimulation. The authors also use c-Trio to demonstrate that indeed LC neurons that project to the LA receive afferents from Orexin cells. The manuscript is clearly written and the methods are sound. However, the authors do not provide causal evidence that the orexin-LC-LA tripartite system is functional; they only demonstrate a behavioral effect on orexin-LC/NA and in separate experiments on LC/NA-LA. Some conclusions cannot be drawn from the original data.

Major concerns:

1) The authors perform optogenetic excitation of LH-LC-LA circuitry to show that they can produce freezing even in an emotionally neutral context. But one key experiment is missing. What happens when optogenetic excitation is given in combination with the tone CS? Does this potentiate the fear response, compared to laser stimulation alone in the absence of the tone cue? This is an important issue that must be addressed.

2) Some of the conclusions are not supported by their findings. For example:

- a. Last sentence of the first results paragraph (Line 110): "animals cannot sustain behavioral expression of fear without the function of OX1R" – this is completely incorrect- actually animals can show- and they exhibit only a minor decrease in freezing time following the removal of OX1R from LC neurons
- b. Lines 179-183: the FOS findings in the TH+ cells do not suggest that orexin neurons are physiologically activated by fear-related cues
- c. Line 183: The authors did not measure NA release – so they cannot reach any conclusions reading this. This sentence should be removed.

3) The paper lacks many essential control groups. For example:

- a. Results presented in Figure 1: the authors lack a control group- the NAT-Cre mice; since without this group the difference in OX1Rf/f and OX1Rf/f::NAT-Cre could be a result of genetic differences between the strains.
- b. Results presented in Figure 2: the authors need to add a control group expressing a control virus, and treated with both saline and CNO.
- c. Results presented in Figure 5: the authors should add a eYFP expressing mice group.

4) All experiments in this manuscript were performed during the light phase- during which the mice are typically asleep. Orexin-LC circuit is known to regulate arousal- thus the results could be attributed to an altered arousal state. It is necessary for at least some of the experiments to be replicated in the dark phase.

5) The effect presented in Fig. 1d is very small – a minor decrease in freezing time only in a single time point that lasted half a minute. Indeed, the effect of the antagonist (Fig. 1g) is much stronger than the local deletion- and suggest that the suggested pathway is not the principle one.

6) The optogenetic stimulation conditions 10 Hz are saturating and possibly inducing artifacts.

7) There is no validation of the mouse line used in this study (OX1Rf/f::NAT-Cre)- to demonstrate he has no general deficits in behavior. The authors should add data regarding his locomotor activity and anxiety related behaviors to demonstrate that this is indeed related to fear memory.

8) The identification of OX-LH neurons that synapse onto LA-projecting LC-NAT cells is extremely sparse. Is this an accurate representation of all such neurons, or is it an underestimation? An important experiment would be to do the non-TRIO modified rabies tracing (from LC-NAT cells) to identify how many SAD-GFP+/OX-LH+ cells are present in general. This will determine whether most of the OX-LH+ cells synapse onto LA-projecting or non-LA-projecting LC-NAT neurons. In other words, what is the significance of this LH-LC-LA circuit if most of the OX-LH cells actually synapse onto the non-LA-projecting LC subset? The simpler modified rabies non-TRIO strategy can address this.

9) 10 mg/kg CNO is a very high dose with significant off-target effects and metabolism to active clozapine etc. Especially without a non-hM4Di CNO-treated control group, I would be skeptical of these results. Please provide the rationale for this dose in Methods, and discuss potential caveats in interpretation of DREADD experiments in the Discussion. On this subject, why was DREADD inhibition rather than vLWO inhibition chosen from the LC manipulations?

10) There is so much classic LH&LC literature to cite, but unfortunately much of it is ignored. Citations should especially be given for studies from Carter et al., which were seminal for causally establishing this LH->LC circuit in physiological arousal. This connection has also been established in zebrafish (Singh et al., *Elife*. 2015 Sep 16; 4:e07000.) and is also ignored. In addition, the Sears et al. study from LeDoux lab is cited, but these results could be compared and contrasted and discussed more deeply. In general, the Discussion is superficial and could be more specific about placing their Results within the context of the literature.

Minor:

The manuscript lacks many references to previous work. For example: line 46, line 49

Lines 53 and 46 are repetitive.

The first time the authors mention NA (line 60) in the text- they should use noradrenergic.

Supplementary figure 2 (line 154) should come before sup 3 (line 92)- change their names and order.

The authors do not explain what is the NAT-Cre line, and should add this info.

Clear, low magnification pictures of OX-cre expression and quantification of double OX-ir should be provided, as current Ox-cre mouse lines are notorious for their ectopic expression.

Line 105- "The same tendency was seen even in the absence of the" – there is NO visible nor statistical significant effect- and this remark should be removed. If the authors would have depicted both the positive and negative SE- it would have been clear that they are overlapping.

Line 107- again, this is a misleading sentence. There is no statistical difference between 0.5-3 min, and the authors should remove this sentence. It is not ok to put a statistical test that examines a separate parameter following a statement.

Figure 1c, 2c, 5e, 5j and all other relevant figures: the authors should perform a two-way repeated measures ANOVA- and not a two-way ANOVA. Please add the p value for the interaction between the two factors.

Why is p-value shown on the panel of fig.1c and not fig.1d?

Fig 2e- the authors should quantify the amount of c-fos in mCherry positive cells and not in TH positive (TH is not a proxy since there seems to be a relatively low co-localization (Fig.2b)). This remark is also relevant to other quantifications of c-fos in the manuscript.

Figure 4- please increase the sample size of the control group to match the experimental group.;

Figure 4f – please change the y axis to 0-20 – so the results will be visible. Please provide details on this manipulation- how long did the stimulation last? Etc.

Line 174- to say that the optogenetic silencing had no effect when the optic fibers were not correctly implanted- you have to compare the data to the CONTROL and not to the other experimental group (Supp fig.4)- please add analysis. ; similar comment to the analysis presented in fig supp 5 and etc.

Fig 5f- please change the y axis to 0-30, and add statistics. Also in the similar figures.
How come the data of the same group Orexin-Cre+AAV-ChR2 mice in fig5e and 5j are so different.
This suggest that the optogenetic stimulation did not work properly in the experiment presented in Fig5j. The authors should repeat this.

Supplementary fig. 3 – the authors also need to report how many neurons are TH- / Tomato positive
Line 188- this sentence is misleading this the authors state that " Given that the LH->LC->LA... might be possible to induce or modulate fear-related behavior by artificially stimulating this circuit" – however – the authors only stimulate the LH->LC circuit. The text should be edited.

Line 195- please provide quantification.

Reviewer #2 (Remarks to the Author):

Overall, I found the premise of the study to be exciting and provocative, as the circuit described is not thought to be a canonical fear circuit. The emphasis on neuropeptides and neuromodulatory systems is important and interesting. Though I am generally positive about this manuscript and believe it has the potential to be an impactful contribution to the field, I have a number of major technical concerns that preclude my ability to support publication of this manuscript in its current form as the data included do not fully support the claims made.

As a note to the journal or the authors, it would be more considerate to the reviewers if the supplemental figures could be merged into a single pdf or at the minimum have file names that indicate figure number.

1. OX1RF/F; NAT-cre mouse: The use of the OX1RxNAT-Cre was an elegant and powerful experiment. However, I need to see some validation that this manipulation works as advertised. First, some validation of the specificity of Cre expression in this mouse is required. Given the concerns that have been raised for a number of other very popular cre lines, quantification of cre-dependent expression and colocalization with an immunohistochemical for NA should be provided. Second, please provide data that show that OX1R are indeed removed, and that this was specific to NAT+ neurons. Thorough high-resolution imaging across multiple brain regions and animals along with quantification is necessary. Third, the LC is not the only brain area expressing NAT. If other NAT-expressing areas also have prominent OX1R activity, the behavioral effects reported in Fig 1D can be mediated by Ox1r-NA circuits other than the LH(OX) - LC(NA). The authors should provide further characterization of the NAT-cre mouse by reporting any cre expression in brain areas other than the LC with Ox1r co-localization. If there is some variability in any of these measures, including the efficacy of the manipulation, it would be nice to see this plotted against the behavior of the animals.

2. Use of the OX1R antagonist SB338467: How did the authors determine the dose used? Did the authors perform a dose-dependent curve? If not, this should be discussed and caveats acknowledged (specifically, at this dose, how does the binding affinity compare to other receptors?)

3. Behavioral paradigm: Across the manuscript, it would be nice to see more controls for the specificity of behavior. Did the authors look at any other measures of behavior? Is there any change in overall locomotion? Most importantly, is this an impairment of fear expression or an impairment of memory retrieval? Given the role of both orexin and noradrenaline in arousal/vigilance/wakefulness, this is actually quite important (see Comment #5).

4. Why did the authors use this extended CS during test day?

5. Along these lines, the authors should discuss this in the context of literature examining LC input to the LA. In general, the scholarship of the manuscript was a bit sloppy and incomplete. Some studies (such as Johansen, J. P. et al. Hebbian and neuromodulatory mechanisms interact to trigger associative memory formation. Proc. Natl. Acad. Sci. 201421304 (2014).

doi:10.1073/pnas.1421304111) were actually cited TWICE (#13 and #20) when other publications that are also related to the study (for example, (Carter et al., 2012, 2013; Siuda et al., 2016) demonstrating the functional role of LH to LC) were not cited.

6. cTRIO disynaptic tracing. The rabies tracing is the most direct anatomical evidence that speaks for the LH-LC-LA disynaptic circuit characterized in this paper. Further quantification of this tracing study will provide greater insight: 1) What proportion of LH(OX) neurons send projections of LC(NA)-LA neurons? 2) What proportion of LC(NA)-LA input come from LH(OX) neurons? Appropriate quantification of this anatomical dataset will be important to put the LH (OX) - LC (NA) - LA disynaptic circuit into a broader systems context.

7. cTRIO data. What about the rest of the brain? There are standard ways of displaying these data across the brain that were not included in the manuscript, and (while I would not require them for this study) would serve as a valuable resource if the authors selected to include them.

8. Broad, non-specific manipulations. Although quantifying Fos expression in the LH, LC and LA with fasting and Ox1R antagonism provides evidence for the involvement of each of these brain areas in mediating the effects of orexin, it does not provide convincing evidence for the role of the specific disynaptic circuit as claimed, especially given the widespread effects of enhanced orexin signaling. Furthermore, administration of SB224867 throughout this manuscript were systemic injections and cannot speak for the specific role in LC (OX1R+ & NA+) — LA circuit.

While the authors were able to show the involvement of individual components (LH(OX)-LC and LC(NA)-LA) of the disynaptic circuit in fear expression, the behavioral effects may be mediated by broad transmission to other brain regions at each intermediate step and do not specifically depend on the LH-LC-LA circuit highlighted. Additional behavioral experiments with more specific manipulations are necessary to clarify the functional relevance of the LH-LC-LA circuit. If the disynaptic LH(OX)-LC(NA)-LA circuit is specifically involved in fear expression, the enhanced freezing observed with excitation of LH(OX) - LC or fasting-induced orexin neuron activation should be eliminated by 1) inhibiting the LC(NA) - LA pathway and 2) local administration of SB224867 into the LC.

9. Optogenetic experiments: Failed fiber placements and injected WT are insufficient controls for optogenetic behavioral experiments. The damage induced by fiber implantation is different if the location of the implant is different, so these are not appropriate controls. Further, in many transgenic cre lines (for example, DAT-Cre mice) there are behavioral phenotypic differences from WT mice, and so wild-type mice are not appropriate controls.

At the minimum, authors should include control groups of NAT-cre/OX1R-cre animals with cre-dependent viral expression of only a fluorophore in the same pathways.

Reviewer #3 (Remarks to the Author):

It is an interesting paper that seeks to understand the important connection between LH-Ox neurons and LA-projecting LC-NA neurons, and fear behaviour, by using pharmacogenetic and optogenetic approaches. Orexin system is also activated physiologically by fasting to show modulation of fear

responses.

Although the premise is interesting and important, there are several serious issues that need to be addressed in order for the claims to match the data:

Regarding the results shown in Fig 1, in 1C there is a worrying decrease of freezing just right after the 4th CS-US. It is important to make sure the mice learn appropriately during the conditioning, and this could indicate that these mice do not learn as well as the controls, compromising the data of the test. Please make comments on this.

In Fig 1G, in the absence of CS, the authors talk about a tendency of lower freezing behaviour although not at statistically significant levels. I do not see this tendency: only 1 point out of 5 is a little bit lower than the control, the rest are the same. And, if something, I would have thought it could be due to less anxiety after the SB injection.

If we look at 1D vs 1G, freezing levels are lower after SB (inhibiting the whole orexin system) than only lacking ORXR in LC NA neurons. If ORXR in LC-NA neurons sustain expression of fear, I would have expected a stronger decrease of freezing in D (ORXR NAT cre mice), or at least similar to those after SB. Please comment.

In general the authors make strong statements based on their results, but the data does not always clearly support the statements:

E.g. when they inhibit the LC NA neurons the authors remark that, during test, the difference is more obvious at the end of the CS presentation, while those mice lacking ORXR1 specifically in LC NA neurons show the opposite, the difference was higher at the beginning of presentation (actually the only one point statistically different, 1D). Then when they silence LH OX -> LC the fear response is also lower in the later phase of CS. I would like them to address and comment on this different fear expression.

In figure 4 they inhibit optogenetically the LH Orx -> LC projection injecting AVV vLWO into the LH of Wt and Orx Cre mice. I think it would be much better to use the same animals (Orx-cre mice) as controls instead of using WT (for example Orx cre-YFP controls?). This happens also in fig 5. It seems viral expression is not controlled for?

It would have been good as well to show the behavioural response of the ORX cre +AVV vLWO mice during CS+ and the laser off. One nice experiment that could be done is to test the behavioural response of these ORX cre +AVV vLWO mice after silencing LH orx -> LC after o/n fasting.

In fig 5E, after the excitation of LH ORX ->LC, the Orx cre AVV CHR2 mice show higher freezing levels without presentation of the cue, almost during all the time the laser is on. In 5J - what are supposed to be the same group of mice (Orx cre AVV ChR2 – vehicle ip n=4) - only show higher fear response at the end of the laser ON, being similar to the SB-injected animals during the first half of the laser on presentation. They are supposed to be the same group of mice (only an ip injection of vehicle as difference), and it is worth mentioning how different their fear responses are. Please comment on this.

In fig 7 they fast WT mice and measure freezing responses to a novel context. I miss the results showing the fear response of fasted animals in both contexts, A and A'. The increase of the freezing could be due mainly to an increase of the anxiety when the orexin system is activated. Introducing the animal to a novel context increases the anxiety of the animals as well. It would be interesting to see how different the fear responses to the cue or the novel context are, and comment on the anxiety due

to novelty.

In general, based on the data presented and the experiments done, I found it too daring to talk so strongly about fear generalization. More experiments focused on testing actual generalization using different CS (-/+), or different context, in the same group of experimental animals should be tested in order to check whether the animals generalize their response to fear in any circumstance, regardless of the cue. Please either perform these experiments or remove claims of generalisation from the ms.

Reviewer #1:

This manuscript reports connectivity between LH orexin neurons, the LC as a major effector and the LA as an output of LC stimulation. The authors also use c-Trio to demonstrate that indeed LC neurons that project to the LA receive afferents from Orexin cells. The manuscript is clearly written and the methods are sound. However, the authors do not provide causal evidence that the orexin-LC-LA tripartite system is functional; they only demonstrate a behavioral effect on orexin-LC/NA and in separate experiments on LC/NA-LA. some conclusions cannot be drawn from the original data.

[Answer]

Although we showed the effects of manipulation on $LH^{\text{orexin}} \rightarrow LC^{\text{NA}}$ and $LC^{\text{NA}} \rightarrow LA$ circuit independently, we believe our work, albeit directly, clearly supports the function of $LH^{\text{orexin}} \rightarrow LC^{\text{NA}} \rightarrow LA$ systems via a combination of several sets of experiments: (i) the existence of the $LH^{\text{orexin}} \rightarrow LC^{\text{NA}} \rightarrow LA$ pathway by cTRIO (Fig. 3); (ii) Excitation of orexin neurons in fearful situations (Fig. S1); (iii) Attenuated freezing behavior by an antagonist for OX1R, which abundantly expressed in LC^{NA} (Fig. 1, 5, 7); (iv) Increase of freezing behavior by fasting, which activates orexin neurons, and inhibition of this effect by an OX1R antagonist, which was locally administered into the LC (Fig. S8). (v) The remarkably similar enhancement of freezing by manipulations on either orexin termini at the LC or NA termini at the LA (Fig. 5, 6); (vi) Decreased freezing in mice in which LC-NA neurons specifically lack expression of OX1R (Fig.1); (vii) We found fasting increased freezing in context A' and, Fos expression in neurons in each component of the $LH^{\text{orexin}} \rightarrow LC^{\text{NA}} \rightarrow LA$ circuit in mice was elevated following fasting, suggesting this circuit is engaged in the increase of freezing time in context A' mimicking the results of optogenetic activation of this circuit. The activations of LC^{NA} neurons as well as LA neurons by fasting were inhibited with local injection of SB3348687 in the LC (Fig. S8). Thus, we feel that this overwhelming amount of convergent data from a diverse set of experiments supports our conclusions as stated in the revised manuscript.

Major concerns:

1) *The authors perform optogenetic excitation of LH-LC-LA circuitry to show that they can produce freezing even in an emotionally neutral context. But one key experiments is missing. What happens when optogenetic excitation is given in combination with the tone CS? Does this potentiate the fear response, compared to laser stimulation alone in the absence of the tone cue? This is an important issue that must be addressed.*

[Answer]

Thank you for pointing out an important issue. According to the reviewer's suggestion, we examined whether the optogenetic stimulation of orexin fibers in the LC could enhance freezing behavior against the CS, which was mildly associated with fear. We used a protocol, in which we presented two sorts of CS. CS1 was 70 dB white noise, which was not associated with foot shock, while CS2 was 2900 Hz, 80 dB tone, which was explicitly associated with foot shock (Fig. 8a). We applied these cues in the home cage, so that the influence by contextual elements should be minimized. In this condition, CS1 only slightly increased freezing time, although it was not statistically significant. We also confirmed that optogenetic stimulation of orexin fibers in the LC alone did not show any effects on freezing time (Fig. 8f). However, by combination of CS1 and optogenetic stimulation of orexin fibers in the LC, robust freezing was evoked (Fig. 8g). This suggests that excitation of orexin terminals in the LC enhanced fear-related behavior against auditory cue, which is mostly ignored in normal situation.

2) *Some of the conclusions are not supported by their findings. For example:*

a. *Last sentence of the first results paragraph (Line 110): "animals cannot sustain behavioral expression of fear without the function of OX1R" – this is completely incorrect- actually animals can show- and they exhibit only a minor decrease in freezing time following the removal of OX1R from LC neurons*

[Answer]

From the result of experiment using OX1R antagonist (Fig. 1g), blockade of OX1R largely decrease later phase of freezing behavior during CS presentation, suggesting that OX1R plays an important role in sustaining freezing. However, we agree that this sentence might cause misleading as the reviewer suggested. We have deleted the sentence "In other words, animals cannot sustain behavioral expression of fear without the function of OX1R."

b. *Lines 179-183: the FOS findings in the TH+ cells do not suggest that orexin neurons are physiologically activated by fear-related cues*

[Answer]

We already showed that orexin neurons are physiologically activated by fear-associated cues in supplementary fig. 1. Because optogenetic silencing of orexin termini in the LC during exposure to conditioned CS caused significantly lower induction of Fos-positive LC-NA neurons as compared with control, we concluded that activated orexin neurons in turn activate downstream LC-NA neurons. To avoid confusion, we modified that sentence,

“suggesting that orexin neurons are physiologically activated by fear-related cues (Supplementary Fig. 1) and, in turn, activate LC-NA neurons” to “Because orexin neurons are physiologically activated by fear-related cues (Supplementary Fig. 1) this activation might lead excitation of LC-NA neurons.” (ll.195-197)

c. Line 183: The authors did not measure NA release – so they cannot reach any conclusions reading this. This sentence should be removed.

[Answer]

We appreciate this feedback and completely agree. According to the reviewer's suggestion, we removed the sentence.

3) The paper lacks many essential control groups. For example:

a. Results presented in Figure 1: the authors lack a control group- the NAT-Cre mice; since without this group the difference in OX1Rf/f and OX1Rf/f::NAT-Cre could be a result of a genetic differences between the strains.

[Answer]

According to the reviewer's suggestion, we conducted the CCF experiments on *NAT-Cre* mice to confirm that these mice show no abnormality in the cued- and contextual fear conditioning as compared with wild-type littermates. We added this data as supplementary fig. 2f-h. These data show that the *NAT-Cre* allele did not affect the phenotype in the CCF, and therefore the difference between *OX1RF/F* and *OX1RF/F; NAT-Cre* presented in Figure 1 does not stem from the existence of *NAT-Cre* allele. We added a sentence “*NAT-Cre* mice with normal *OX1R* alleles did not show any abnormality in the cued- and contextual fear test as compared with wild type littermates, suggesting that existence of *NAT-Cre* allele did not affect the phenotype (Supplementary Figs. 2f-h).” (ll.102-104)

b. Results presented in Figure 2: the authors need to add a control group expressing a control virus, and treated with both saline and CNO.

[Answer]

We acknowledge the importance of such control groups, however, very large numbers of mice would be sacrificed if we proceed such experiments. Considering the “3Rs” principle, we don't think it is appropriate to examine control virus and treatment with both saline and CNO. Further, this is simply not the standard in the field. Many published works using the DREADD system do not show such control groups (eg. Kunio Kondo et al., Nature, 2016, Yu Hayashi et al., Science, 2015, etx). In addition, as mentioned above, since we confirmed

there was no significant difference between WT (*NAT-Cre-*) and *NAT-Cre+* in cued and contextual fear conditioning tests, we believe these results are sufficient to provide the evidence that shows the effects were specific.

c. Results presented in Figure 5: the authors should add a eYFP expressing mice group.

[Answer]

We performed an additional experiment using different CS (CS1, white noise and CS2, tone) combined with optogenetic stimulation to examine whether $LH^{Orexin} \rightarrow LC$ stimulation potentiates cued fear generalization (Fig. 8). In this experiment, we injected *AAV-EF1a-DIO-EYFP* or *AAV-EF1a-DIO-ChR2* into the LH^{Orexin} neurons in *Orexin-Cre* mice and illuminated at LC. In this experiment, we confirmed clear difference between EYFP and ChR2 groups, and EYFP expressing group did not show any effect by laser stimulation. In addition, we have already shown that ChR2 injected Cre-negative group did not show any effect for laser stimulation (Fig. 4, 5, 6), we think the effect is specifically observed when ChR2 is expressed in orexin neurons.

4) All experiments in this manuscript were performed during the light phase- during which the mice are typically asleep. Orexin-LC circuit is known to regulate arousal- thus the results could be attributed to an altered arousal state. It is necessary for at least some of the experiments to be replicated in the dark phase.

[Answer]

Historically most behavioral experiments including CCF experiments have been done in the light phase. Usually mice stay awake with high vigilant state in the experimental condition even in the light period. This vigilance level is further enhanced with behavioral testing in a novel environment; thus, mice invariably stay awake. Since we examined the fear-related behavior, not sleep/wakefulness states or biological clock, we don't feel these experiments are necessary to support our conclusions.

5) The effect presented in Fig. 1d is very small – a minor decrease in freezing time only in a single time point that lasted half a minute. Indeed, the effect of the antagonist (Fig. 1 g) is much stronger than the local deletion- and suggest that the suggested pathway is not the principle one.

[Answer]

We respectfully disagree with the reviewer's interpretation of the statistical results. ANOVA analysis demonstrated the difference between two groups was highly significant, although strict post-hoc analysis suggested the pairwise significance in only one point. We have

modified the figs to indicate p values.

However, to address the reviewer's concerns, we added a sentence "Although we could not compare result from two separated experiments, the effect of antagonist might be larger as compared with local deletion of OX1R in LC^{NA} neurons (Figure 1g), suggesting that other pathway(s) involving OX1R signaling might play an additional role." (ll. 370-373).

6) *The optogenetic stimulation conditions 10 Hz are saturating and possibly inducing artifacts.*

[Answer]

Many existing papers used the same or similar condition with ours in the similar experiments, thus we feel we are well in line with the standards of the field. We think 10 Hz is appropriate as previous study already used that frequency for the stimulation of LC^{NA} neurons (Jordan G. McCall et al., Neuron, 2015).

7) *There is no validation of the mouse line used in this study (OX1R^{f/f}::NAT-Cre)- to demonstrate he has no general deficits in behavior. The authors should add data regarding his locomotor activity and anxiety related behaviors to demonstrate that this is indeed related to fear memory.*

[Answer]

We thank the reviewer for the suggestion. Although further experiments would be ideal to characterize the behavior, since we had already found that global deletion of OX1R lead to slight abnormalities in locomotor activity and anxiety related behaviors (Front Behav Neurosci. 2015 Dec 10;9:324. doi: 10.3389/fnbeh.2015.00324.), they are beyond the scope of this study, because we focused on the function in the cued- and contextual fear memory itself here. We are planning to do comprehensive behavioral study on these mice and report the results in somewhere else.

8) *The identification of OX-LH neurons that synapse onto LA-projecting LC-NAT cells is extremely sparse. Is this an accurate representation of all such neurons, or is it an underestimation? An important experiment would be to do the non-TRIO modified rabies tracing (from LC-NAT cells) to identify how many SAD-GFP+/OX-LH+ cells are present in general. This will determine whether most of the OX-LH+ cells synapse onto LA-projecting or non-LA-projecting LC-NAT neurons. In other words, what is the significance of this LH-LC-LA circuit if most of the OX-LH cells actually synapse onto the non-LA-projecting LC subset? The simpler modified rabies non-TRIO strategy can address this.*

[Answer]

SADΔG is effectively transported from post-synaptic neurons to pre-synaptic neurons only when the synapses between these two neurons are tight synapses. When neurons make loose synapse, and utilize volume transmission, these neurons are not likely to be depicted. For example, we did not find any positive cells in the dorsal raphe nucleus by retrograde tracing of orexin neurons (our unpublished result), although another study using ChR2 as an anterograde tracer and an optogenetic tool, clearly suggested that dorsal raphe neurons send functional input to orexin neurons. Since peptidergic neurons also largely make loose synapses, although they sometimes tight ones, the tracing efficiency of these cells might be way lower than that of Glutamatergic/GABAergic neurons. This is a reason why LH^{orexin} neurons that synapse onto LA-projecting LC^{NA} cells seem to be sparse. Therefore, we think the number was underestimated.

We added following sentences “In this study, we employed cTRIO to confirm that LH^{orexin} neurons make direct synaptic contact with LC^{NA} neurons that send innervations to the LA (Fig. 3), suggesting the existence of a tripartite LH^{orexin}→LC^{NA}→LA pathway. Notably, SADΔG is likely to be effectively transported from post-synaptic neurons to pre-synaptic neurons only when the synapses between these two neurons are classical, tight synapses. When neurons make loose synapses and/or utilize volume transmission, these neurons are not likely to be detected by the rabies virus. Although peptidergic neurons can form tight synapses, the majority are thought to be loose synapses, thus we speculate that the number of labeled orexin neurons in our cTRIO experiment (Fig. 3) might be underestimate of the actual connectivity.” (ll.384-392)

9) 10 mg/kg CNO is a very high dose with significant off-target effects and metabolism to active clozapine etc. Especially without a non-hM4Di CNO-treated control group, I would be skeptical of these results. Please provide the rationale for this dose in Methods, and discuss potential caveats in interpretation of DREADD experiments in the Discussion. On this subject, why was DREADD inhibition rather than vLWO inhibition chosen from the LC manipulations?

[Answer] First, we are sorry for confusion, we made a mistake in the calculation of the working concentration and the actual dose we used was 5 mg/kg. In many studies CNO has been used to acutely activate DREADDs with CNO doses up to 20 mg/kg without any evidence of side effects (Smith et al. Behav Neurosci. 2016;130:137-55). Because we need maximal inhibition of LC^{NA} neurons, we selected the 5 mg/kg dose here. We added the information in drugs section in online methods (ll.45-46). Also, since it was difficult to optogenetically excite orexin terminals in the LC with simultaneous inhibition of LC^{NA}

neuronal somata, we employed DREADD inhibition.

10) There is so much classic LH&LC literature to cite, but unfortunately much of it is ignored. Citations should especially be given for studies from Carter et al., which were seminal for causally establishing this LH->LC circuit in physiological arousal. This connection has also been established in zebrafish (Singh et al., Elife. 2015 Sep 16;4:e07000.) and is also ignored. In addition, the Sears et al. study from LeDoux lab is cited, but these results could be compared and contrasted and discussed more deeply. In general, the Discussion is superficial and could be more specific about placing their Results within the context of the literature.

[Answer]

Thank you for pointing out the lack of citation, we apologize for the oversight. We modified the discussion to cite these studies in the revised manuscript. Also, we added more detailed discussion of our data in relation to the Sears et al work in our discussion.

Minor:

The manuscript lacks many references to previous work. For example: line 46, line 49

Lines 53 and 46 are repetitive.

The first time the authors mention NA (line 60) in the text- they should use noradrenergic.

Supplementary figure 2 (line 154) should come before sup 3 (line 92)- change their names and order.

The authors do not explain what is the NAT-Cre line, and should add this info.

[Answer]

Thank you for your suggestion. We fixed these mistakes and added references and information of *NAT-Cre* mice in 'Animals' section of the online methods (ll. 5-9).

Clear, low magnification pictures of OX-cre expression and quantification of double OX-ir should be provided, as current Ox-cre mouse lines are notorious for their ectopic expression.

[Answer]

We already showed low and high magnification pictures of OX+/vLWO+ or OX+/ChR2+ cells (Fig. 4c, 5c). We added percentage of OX+/vLWO+ or OX+/ChR2+ cell ratio in the results section of the manuscript. (vLWO; ll.177-178, ChR2; ll.201-204).

Line 105- "The same tendency was seen even in the absence of the" – there is NO visible

nor statistical significant effect- and this remark should be removed. If the authors would have depicted both the positive and negative SE- it would have been clear that they are overlapping.

[Answer]

According to the reviewer's suggestion, we deleted the sentence.

Line 107- again, this is a misleading sentence. There is no statistical difference between 0.5-3 min, and the authors should remove this sentence. It is not ok to put a statistical test that examines a separate parameter following a statement.

[Answer] We apologize for our lack of clarity. In this sentence, we meant time window of 3.5-5 min, which corresponds to 0.5-3 min after the start of the CS. As shown in Fig. 1g, we found a statistically significant difference in each time point in 3.5-5 min. We modified the sentence "The effect of the antagonist was not evident for first 30 s of CS presentation, but decreased freezing behavior for the subsequent period (unpaired two-tailed Student's *t*-test, $t = 4.411$, $p = 0.0002$, Fig. 1g)" to "The effect of the antagonist was not evident for first 30 s of CS presentation, but we observed dramatic reduction in freezing behavior for the remainder of the tone (Two-Way RM ANOVA with Sidak's post-hoc test, $F_{(1, 24)} = 11.47$, $p = 0.0024$, 3.5-5 min in Fig. 1g-left)" (ll.112-115).

Figure 1c, 2c, 5e, 5j and all other relevant figures: the authors should perform a two-way repeated measures ANOVA- and not a two-way ANOVA. Please add the p value for the interaction between the two factors.

Why is p-value shown on the panel of fig.1c and not fig.1d?.

[Answer]

We re-analyzed the data of Fig. 1c, 2c, 5e, 5j and all other relevant figures with Two-Way RM measures ANOVA. We also deleted p-value shown on the fig. 1c, 1f, 2c, 4d, 5d, 5i, 6d, 7c and added these information in result section of the manuscript.

Fig 2e- the authors should quantify the amount of c-fos in mCherry positive cells and not in TH positive (TH is not a proxy since there seems to be a relatively low co-localization (Fig.2b)). This remark is also relevant to other quantifications of c-fos in the manuscript.

[Answer]

According to the reviewer's suggestion, we quantified Fos in mCherry positive cells and showed the data as Fig. 2g. We added following sentences "After the behavioral tests, we confirmed that the number of Fos- and TH-double labeled cells, Fos- and mCherry-double

labeled cells in the LC were significantly reduced by CNO injection [NAT-Cre+;hM4Di (Saline), $n = 3$; NAT-Cre+;hM4Di (CNO), $n = 3$: $t = 9.309$, $p = 0.0024$, Fig. 2e, f] [NAT-Cre+;hM4Di (Saline), $n = 4$; NAT-Cre+;hM4Di (CNO), $n = 4$: $t = 3.404$, $p = 0.0283$, Fig. 2g]". (ll.138-142).

Figure 4- please increase the sample size of the control group to match the experimental group.;

[Answer]

We used clear statistical tests and applied reasonable thresholds. Compared with some other studies in this field it seems that this one is probably reasonably well powered. More subjects are always better but statistical tests provide some security when making inferences based on smaller sample sizes, which allow us to limit the numbers of animals sacrificed for the experiments.

Figure 4f – please change the y axis to 0-20 – so the results will be visible. Please provide details on this manipulation- how long did the stimulation last? Etc

[Answer].

According to the reviewer's suggestion, we changed the y axis to 0-20, and add changed the sentence "Auditory CS or laser stimulation itself before the conditioning had no effect on freezing behavior" to "Auditory CS or laser stimulation with the same protocol with Fig 4e without prior conditioning had no effect on freezing behavior" (ll.186-187).

Line 174- to say that the optogenetic silencing had no effect when the optic fibers were not correctly implanted- you have to compare the data to the CONTROL and not to the other experimental group (Supp fig.4)- please add analysis. ; similar comment to the analysis presented in fig supp 5 and etc.

[Answer]

We changed dataset and compare the failed implantation group to the control. We showed changed figures in Supplementary Fig. 4-6.

Fig 5f- please change the y axis to 0-30, and add statistics. Also in the similar figures.

[Answer]

We changed y axis to 0-30 in Fig. 5f, k and wrote statistics in result section of the manuscript.

How come the data of the same group Orexin-Cre+AAV-ChR2 mice in fig5e and 5j are so different. This suggest that the optogenetic stimulation did not work properly in the experiment presented in Fig5j. The authors should repeat this.

[Answer]

Since optogenetic stimulation exhibited significant increase in time of freezing in both experiments, we think it works properly (Fig. 5j). Also, these groups were not the same, we used different sample in each experiment. We agree that temporal patterns of increase in freezing seem to be somewhat different from each other in those experiments, but qualitatively, both experiments clearly showed increase of freezing times by optogenetic stimulation.

Supplementary fig. 3 – the authors also need to report how many neurons are TH- / Tomato positive

[Answer]

According to the reviewer's suggestion, we counted TH-/Td-tomato positive cells in the LC and found that to be $9.39 \pm 1.1\%$. Please confirm supplementary legend Fig. 2 (ll.18-20).

Line 188- this sentence is misleading this the authors state that “ Given that the LH->LC->LA... might be possible to induce or modulate fear-related behavior by artificially stimulating this circuit” – however – the authors only stimulate the LH->LC circuit. The text should be edited.

[Answer]

We also stimulated downstream $LC^{NA} \rightarrow LA$ circuit and confirmed increased freezing response in a context A' same as the stimulation of upstream $LH^{orexin} \rightarrow LC$ in Fig. 5 (Fig. 6). Therefore we suggests that $LH^{orexin} \rightarrow LC^{NA} \rightarrow LA$ circuit mediates fear-related behavior. However, to avoid confusions, we modified the sentence to “Given that the $LH^{orexin} \rightarrow LC$ pathway seems to be recruited during behavioral fear expression and plays a role in enhancing and sustaining freezing behavior, ...” (ll.198-199)

Line 195- please provide quantification.

[Answer]

We added percentage of LH^{orexin} neurons expressing ChR2 in result section of the manuscript (ll.201-204).

Reviewer #2 (Remarks to the Author):

Overall, I found the premise of the study to be exciting and provocative, as the circuit described is not thought to be a canonical fear circuit. The emphasis on neuropeptides and neuromodulatory systems is important and interesting. Though I am generally positive about this manuscript and believe it has the potential to be an impactful contribution to the field, I have a number of major technical concerns that preclude my ability to support publication of this manuscript in its current form as the data included do not fully support the claims made.

As a note to the journal or the authors, it would be more considerate to the reviewers if the supplemental figures could be merged into a single pdf or at the minimum have file names that indicate figure number.

[Answer]

Authors thank the reviewer for his/her positive comments and valuable suggestions. According to the reviewer's suggestion, we integrated supplementary figures to a single pdf.

1. OX1RF/F; NAT-cre mouse: The use of the OX1RxNAT-Cre was an elegant and powerful experiment. However, I need to see some validation that this manipulation works as advertised. First, some validation of the specificity of Cre expression in this mouse is required. Given the concerns that have been raised for a number of other very popular cre lines, quantification of cre-dependent expression and colocalization with an immunohistochemical for NA should be provided.

Second, please provide data that show that OX1R are indeed removed, and that this was specific to NAT+ neurons. Thorough high-resolution imaging across multiple brain regions and animals along with quantification is necessary.

Third, the LC is not the only brain area expressing NAT. If other NAT-expressing areas also have prominent OX1R activity, the behavioral effects reported in Fig 1D can be mediated by Ox1r-NA circuits other than the LH(OX) - LC(NA). The authors should provide further characterization of the NAT-cre mouse by reporting any cre expression in brain areas other than the LC with Ox1r co-localization. If there is some variability in any of these measures, including the efficacy of the manipulation, it would be nice to see this plotted against the behavior of the animals.

[Answer]

In addition to the figure 6c that shows specific ChR2 expression in LC^{NA} neurons, we added the percentage of LC^{NA} neurons which expressed ChR2 in the result section of the

manuscript (ll.253-254). We confirmed specific deletion of *OX1R* mRNA in LC^{NA} neurons with combination of *in situ* hybridization and immunohistochemistry for detecting *OX1R* and TH (Supplementary Fig. 2d). In *NAT-Cre* mice, we could not see any strong expression of Cre recombinase in other area except the LC by observing the Cre activity in reporter mice (Supplementary Fig. 2b). Therefore, it is hard to compare the effect of *OX1R* deletion in other area.

2. Use of the *OX1R* antagonist SB338467: How did the authors determine the dose used? Did the authors perform a dose-dependent curve? If not, this should be discussed and caveats acknowledged (specifically, at this dose, how does the binding affinity compare to other receptors?)

[Answer]

Dose dependent effect of SB338467 against cued fear conditioning is already shown in previous study, which we cited in this manuscript (Flores et al., *Neuropsychopharmacology*, 2014). They performed 3 mg-, 5 mg-, 10 mg/kg for I.P injection, and significant effect compared to the vehicle was already seen in 5 mg/kg condition. Therefore, we decided to use 5 mg/kg for our experiment.

3. Behavioral paradigm: Across the manuscript, it would be nice to see more controls for the specificity of behavior. Did the authors look at any other measures of behavior? Is there any change in overall locomotion? Most importantly, is this an impairment of fear expression or an impairment of memory retrieval? Given the role of both orexin and noradrenaline in arousal/vigilance/wakefulness, actually this is quite important (see Comment #5).

[Answer]

Unfortunately, we didn't measure any other behaviors, including locomotor activity, because these were outside of the scope of our work. In this manuscript we explicitly focused on the role of LH^{orexin}→LC^{NA}→LA circuit focusing on freezing behavior. Of course, we acknowledge the importance of other behavioral abnormality, and we are currently analyzing the line with various behavioral paradigms, and hope to publish our findings in the future. In the present study, we mainly focused on the function of orexin pathway in emotional memory retrieval. Since administration of antagonist just before the test session, or inhibitory manipulation of orexin terminal during the test session significantly decreased freezing, it is reasonable to speculate that this is an impairment of fear expression. It is very hard to distinguish the impairment of fear expression and memory retrieval. However, considering the observation that optogenetic excitation enhanced freezing only when animals showed small level of

freezing beyond normal level (Fig. S9), that excitation of LH-LC pathway seems to enhance behavioral expression of fear.

4. Why did the authors use this extended CS during test day?

[Answer]

Considering the slow kinetics of peptidergic and monoaminergic transmission, we thought it was better to use more wider time window to precisely evaluate the contribution of these transmissions in cue-induced freezing behavior.

5. Along these lines, the authors should discuss this in the context of literature examining LC input to the LA. In general, the scholarship of the manuscript was a bit sloppy and incomplete. Some studies (such as Johansen, J. P. et al. Hebbian and neuromodulatory mechanisms interact to trigger associative memory formation. Proc. Natl. Acad. Sci. 201421304 (2014). doi:10.1073/pnas.1421304111) were actually cited TWICE (#13 and #20) when other publications that are also related to the study (for example, (Carter et al., 2012, 2013; Siuda et al., 2016) demonstrating the functional role of LH to LC) were not cited.

[Answer]

According to the reviewer's critical suggestion, we deleted the duplicated citation and cited studies which is pointed out.

6. cTRIO disynaptic tracing. The rabies tracing is the most direct anatomical evidence that speaks for the LH-LC-LA disynaptic circuit characterized in this paper. Further quantification of this tracing study will provide greater insight: 1) What proportion of LH(OX) neurons send projections of LC(NA)-LA neurons? 2) What proportion of LC(NA)-LA input come from LH(OX) neurons? Appropriate quantification of this anatomical dataset will be important to put the LH (OX) - LC (NA) - LA disynaptic circuit into a broader systems context.

[Answer]

SAD Δ G is likely to be effectively transported from post-synaptic neurons to pre-synaptic neurons only when the synapses between these two neurons are tight synapses. When neurons make loose synapse, and utilize volume transmission, these neurons are not likely to be depicted. For example, we did not find any positive cells in the dorsal raphe nucleus by retrograde tracing of orexin neurons (our unpublished result), although another study using ChR2 as an anterograde tracer and an optogenetic tool, clearly suggested that dorsal raphe

neurons send functional input to orexin neurons. Since peptidergic neurons also largely make loose synapses, although they sometimes tight ones, the tracing efficiency of these cells might be way lower than that of glutamatergic/GABAergic neurons. This is a reason why LH^{orexin} neurons that synapse onto LA-projecting LC^{NA} cells are sparse. Therefore, quantifying the population of input cells in this experiment using cTRIO is not considered to be appropriate for understanding these neural circuit. Also, as shown in Fig. 3e, 2.78 % of LC^{NA} neurons directly connected to LA-projecting LC^{NA} neurons. It is possible that these neurons might be mediated by LH^{orexin} neurons and indirectly activate LA-projecting LC^{NA} neurons.

We added following sentences “In this study, we employed cTRIO to confirm that LH^{orexin} neurons make direct synaptic contact with LC^{NA} neurons that send innervations to the LA (Fig. 3), suggesting the existence of a tripartite LH^{orexin}→LC^{NA}→LA pathway. Notably, SADΔG is likely to be effectively transported from post-synaptic neurons to pre-synaptic neurons only when the synapses between these two neurons are classical, tight synapses. When neurons make loose synapses and/or utilize volume transmission, these neurons are not likely to be detected by the rabies virus. Although peptidergic neurons can form tight synapses, the majority are thought to be loose synapses, thus we speculate that the number of labeled orexin neurons in our cTRIO experiment (Fig. 3) might be underestimate of the actual connectivity.” (ll.382-390).

7. cTRIO data. What about the rest of the brain? There are standard ways of displaying these data across the brain that were not included in the manuscript, and (while I would not require them for this study) would serve as a valuable resource if the authors selected to include them.

[Answer]

We had already shown SAD Δ G-EGFP positive cells in other regions including VM (Ventral Medulla), p1PAG(p1 periaqueductal gray), LH (Lateral hypothalamus), MPB(Medial parabrachial nucleus), CeA (Central amygdala), PVN (Paraventricular nucleus) in Supplementary Fig. 3c.

8. Broad, non-specific manipulations. Although quantifying Fos expression in the LH, LC and LA with fasting and Ox1R antagonism provides evidence for the involvement of each of these brain areas in mediating the effects of orexin, it does not provide convincing evidence for the role of the specific disynaptic circuit as claimed, especially given the widespread effects of enhanced orexin signaling. Furthermore, administration of SB224867 throughout

this manuscript were systemic injections and cannot speak for the specific role in LC (OX1R+ & NA+) — LA circuit.

[Answer]

We did the same experiment with microinjection of SB334867 into the LC and got same result as I.P injection (Supplementary Fig. 8). We believe this study further confirm the importance of OX1R in the LC in the emergence of fear-related behavior.

While the authors were able to show the involvement of individual components (LH(OX)-LC and LC(NA)-LA) of the disynaptic circuit in fear expression, the behavioral effects may be mediated by broad transmission to other brain regions at each intermediate step and do not specifically depend on the LH-LC-LA circuit highlighted. Additional behavioral experiments with more specific manipulations are necessary to clarify the functional relevance of the LH-LC-LA circuit. If the disynaptic LH(OX)-LC(NA)-LA circuit is specifically involved in fear expression, the enhanced freezing observed with excitation of LH(OX) - LC or fasting-induced orexin neuron activation should be eliminated by 1) inhibiting the LC(NA) - LA pathway and 2) local administration of SB224867 into the LC.

[Answer]

According to the reviewer's suggestion, we performed the fasting experiment with bilateral microinjection of SB334867 into the LC (Supplementary Fig. 8). After cued fear conditioning followed with 16 h fasting, we bilaterally injected SB334867 into the LC and put the mice into similar but distinct context (A'). As we expected, we confirmed fasting-induced freezing was significantly reduced with SB334867 delivery suggesting that we could partly prove disynaptic LH^{orexin}→LC^{NA}→LA circuit is specifically involved in fear expression.

9. Optogenetic experiments: Failed fiber placements and injected WT are insufficient controls for optogenetic behavioral experiments. The damage induced by fiber implantation is different if the location of the implant is different, so these are not appropriate controls. Further, in many transgenic cre lines (for example, DAT-Cre mice) there are behavioral phenotypic differences from WT mice, and so wild-type mice are not appropriate controls. At the minimum, authors should include control groups of NAT-cre/OX1R-cre animals with cre-dependent viral expression of only a fluorophore in the same pathways.

[Answer]

We performed additional experiments using different CS (CS1, white noise and CS2, tone) combined with optogenetic stimulation to examine whether LH^{orexin}→LC stimulation potentiates cued fear generalization shown in Fig. 8. In this experiment, we injected control virus AAV-EF1a-DIO-EYFP or AAV-EF1a-DIO-ChR2 into LH^{orexin} neurons in *Orexin-Cre*

mice and illuminated at LC. In this experiment, we confirmed clear difference in effect evoked by laser manipulation between EYFP and ChR2 groups.

Reviewer #3 (Remarks to the Author):

It is an interesting paper that seeks to understand the important connection between LH-Ox neurons and LA-projecting LC-NA neurons, and fear behaviour, by using pharmacogenetic and optogenetic approaches. Orexin system is also activated physiologically by fasting to show modulation of fear responses.

Although the premise is interesting and important, there are several serious issues that need to be addressed in order for the claims to match the data:

Regarding the results shown in Fig 1, in 1C there is a worrying decrease of freezing just right after the 4th CS-US. It is important to make sure the mice learn appropriately during the conditioning, and this could indicate that these mice do not learn as well as the controls, compromising the data of the test. Please make comments on this.

[Answer]

Thank you for highlighting this very important point. Since these mice are likely to show abnormality in fear expression itself, it is important to assess if the learning process might be also impaired. However, we found no statistical difference between control and conditional deletion mice with Two-Way RM ANOVA ($OX1R^{F/F}$, $n = 8$; $OX1R^{F/F}; NAT-Cre$, $n = 12$: Two-Way RM ANOVA with Sidak's post-hoc test, $F_{(1, 18)} = 2.673$, $p = 0.1194$, Fig. 1c) (I197-99). Also, the percentage of freezing during 5th CS-US showed no clear difference, thus it is difficult to conclude that these mice could not learn as well as the controls.

In Fig 1G, in the absence of CS, the authors talk about a tendency of lower freezing behaviour although not at statistically significant levels. I do not see this tendency: only 1 point out of 5 is a little bit lower than the control, the rest are the same. And, if something, I would have thought it could be due to less anxiety after the SB injection.

[Answer]

According to the reviewer's suggestion, we deleted the sentence.

If we look at 1D vs 1G, freezing levels are lower after SB (inhibiting the whole orexin system)

than only lacking ORXR in LC NA neurons. If ORXR in LC-NA neurons sustain expression of fear, I would have expected a stronger decrease of freezing in D (ORXR NAT cre mice), or at least similar to those after SB. Please comment.

[Answer]

Although it is difficult to compare the result of two separate sets of experiments, we agree that there is a possibility that *OX1RF/F;NAT-Cre* mice show milder phenotype as compared with mice with acute blockade of OX1R signaling. These mice might be undergoing chronic plastic alterations in the LC, as we previously showed in *orexin/ataxin-3* mice (Tsuji et al. 2013, <https://doi.org/10.1371/journal.pone.0070012>), which might modify the phenotype. Alternatively, since OX1R is also expressed in many other brain regions, global blockade of OX1R signals might result in stronger effect. Our previous study showed global deletion of OX1R (OX1RKO mice) exhibited severe phenotype, supporting the latter possibility.

We added following sentence in the discussion.

“Indeed, acute blockade of OX1R, a receptor abundantly expressed in LC^{NA} neurons, resulted in a significant reduction of CS-mediated freezing. Although we could not compare result from two separated experiments, the effect of the antagonist seems to be larger as compared with local deletion of OX1R in LC^{NA} neurons (Fig. 1g), suggesting that another pathway(s) involving OX1R signaling might play an additional role.” (ll.367-371).

In general the authors make strong statements based on their results, but the data does not always clearly support the statements:

E.g. when they inhibit the LC NA neurons the authors remark that, during test, the difference is more obvious at the end of the CS presentation, while those mice lacking ORXR1 specifically in LC NA neurons show the opposite, the difference was higher at the beginning of presentation (actually the only one point statistically different, 1D). Then when they silence LH OX -> LC the fear response is also lower in the later phase of CS. I would like them to address and comment on this different fear expression.

[Answer]

Thank you for the suggestion. We think it is difficult to compare the effect of genetic deletion and pharmacological acute blockade, because genetic deletion always includes compensatory effect. However, as shown in Fig. 1d and 1g, both groups showed differences after 3.5 min as compared with controls, while we did not see any effects in 3 min point. Thus, we postulated that OX1R plays an important role in maintenance of freezing behavior.

In figure 4 they inhibit optogenetically the LH Orx -> LC projection injecting AVV vLWO into

the LH of Wt and Orx Cre mice. I think it would be much better to use the same animals (Orx-cre mice) as controls instead of using WT (for example Orx cre-YFP controls?). This happens also in fig 5. It seems viral expression is not controlled for?

[Answer]

Thank you for the suggestion. It is ideal to add such a control experiment. Unfortunately, due to the limitations of time and a shortage of mice we could not perform the additional experiments. However, *Orexin-Cre* mice have already been widely used for various kind of experiments and we could not see any behavioral abnormality. We think it is sufficient to use WT(*Orexin-Cre*-) as a control with the same virus injection as has been done in a previous study (Tomoko Isosaka et al., *Cell*, 2015). Additionally, we used viral control (EYFP) in additional experiments shown in Fig. 8.

It would have been good as well to show the behavioural response of the ORX cre +AVV vLWO mice during CS+ and the laser off. One nice experiment that could be done is to test the behavioural response of these ORX cre +AVV vLWO mice after silencing LH orx -> LC after o/n fasting.

[Answer]

Thank you for the suggestion. Although shortage of time and animal resources do not allow us to perform these experiments, we added a fasting experiment with microinjection of SB334867 into the LC (Supplementary Fig. 8) instead of optogenetic silencing. This manipulation also showed the same effect to the global inhibition of OX1R with I.P injection of SB334867, suggesting that fasting-induced freezing in context A' could be silenced with OX1R signaling in the LC.

In fig 5E, after the excitation of LH ORX ->LC, the Orx cre AVV CHR2 mice show higher freezing levels without presentation of the cue, almost during all the time the laser is on. In 5J - what are supposed to be the same group of mice (Orx cre AVV ChR2 – vehicle ip n=4) - only show higher fear response at the end of the laser ON, being similar to the SB-injected animals during the first half of the laser on presentation. They are supposed to be the same group of mice (only an ip injection of vehicle as difference), and it is worth mentioning how different their fear responses are. Please comment on this.

[Answer]

Although we agree with you that the temporal patterns of response are a little bit different from each other, it is difficult to compare two separate experiments. These groups showed

different patterns even in conditioning period. Considering the size of individual differences, to increase numbers of experiments would make them similar patterns, but judging from broader basis, both groups show similar patterns. Since we did not find any statistical difference between these groups, we did not include it in the text or discussion.

In fig 7 they fast WT mice and measure freezing responses to a novel context. I miss the results showing the fear response of fasted animals in both contexts, A and A'. The increase of the freezing could be due mainly to an increase of the anxiety when the orexin system is activated. Introducing the animal to a novel context increases the anxiety of the animals as well. It would be interesting to see how different the fear responses to the cue or the novel context are, and comment on the anxiety due to novelty.

[Answer]

If we put these mice in context A, it evokes very large freezing, masking the effect of fasting (ceiling effect). We thank you for suggesting the possible effect of anxiety. In this study we focused on fear, rather than anxiety. Anxiety should be examined by different paradigms such as EPM, or open field tests. We are going to perform such experiments, and hope to report these data in future publications. However, we have added the text concerning anxiety in final paragraph of the Discussion.

In general, based on the data presented and the experiments done, I found it too daring to talk so strongly about fear generalization. More experiments focused on testing actual generalization using different CS (-/+), or different context, in the same group of experimental animals should be tested in order to check whether the animals generalize their response to fear in any circumstance, regardless of the cue. Please either perform these experiments or remove claims of generalization from the ms.

[Answer]

Thank you for suggesting important point. We performed another experiment using different CS (CS1, white noise and CS2, tone) combined with optogenetic stimulation, to examine whether $LH^{orexin} \rightarrow LC$ stimulation potentiates cued fear expression (Fig. 8). First, we applied different CS (only CS2 was paired with US) in conditioning period (Fig. 8d). During conditioning, CS-induced freezing was different between CS1 and CS2. Next, we optogenetically stimulated $LH^{orexin} \rightarrow LC$ pathway in homecage condition to exclude contextual effect. Optogenetic stimulation didn't show any increase of freezing time compared with the Fig. 5, 6 (Fig. 8f). Otherwise, this stimulation significantly potentiated CS1 induced mild freezing in homecage (Fig. 8g) suggesting that activation of $LH^{orexin} \rightarrow$

LC^{NA}→LA pathway potentiates fear expression and evoke unusual fear response against otherwise ignored auditory cue. Considering the effect we showed in Fig. 5, 6, we assume that LH^{orexin}→LC^{NA}→LA pathway potentiates cue and contextual including elements that are mildly associated with aversive stimulus and mimic fear generalization.

Reviewers' comments:

Reviewer #1 (Remarks to the Author):

The revised manuscript is much improved with the addition of several key experiments, including paired orexin fiber stimulation in LC with a CS, controls including contextual fear in NAT-cre.

Relatively minor concerns remain, including the choice of only one optogenetic stimulation frequency in LC (10 Hz) that is much higher than the spontaneous firing frequency in LC neurons (2-5 Hz) (Aston Jones et al., 1981). Also, Carter et al 2010 used optogenetic stimulation frequencies 1-10 Hz and frequencies >5 Hz appear to be saturating.

Also, controls for CNO are necessary because this compound affects behavior under certain conditions and concentrations (Mc Claren et al., eNeuro 2016).

Reviewer #2 (Remarks to the Author):

The authors have done a moderate job of addressing reviewer concerns. Some of my concerns were addressed, however there are still two major technical concerns remaining that preclude publication in my opinion.

Multiple reviewers raised a concern about the lack of causal evidence that the orexin-LC-LA system is functional, since the authors only provide info about orexin-LC/NA and then LC/NA-LA. The authors respond that they "believe their work, albeit directly, clearly supports..." Do they mean indirectly? They don't show that the same neurons that are receiving orexin input in LC are the ones projecting to LA (in a functional manner, only with cTRIO).

In response to point 3b raised by reviewer 1, the authors refuse to do the appropriate controls. While I might accept the argument that a control virus group need not have both saline or CNO (though this would be optimal), they MUST have a CNO control group. There have been a number of studies raising the issue, including <https://www.ncbi.nlm.nih.gov/pmc/articles/PMC5089539/>.

These reports (that CNO on its own has effects on startle behavior) and alters catecholaminergic tone. This is a major confound that the authors must control for. I am shocked by their response to refuse to do such critical controls in the face of evidence suggesting nonspecific effects of CNO. Citing studies that failed to the appropriate controls (that were accepted/published before these new concerns were raised in the community) is not acceptable.

To the journal/authors: Despite my request to label the figures in the first round, this request was ignored and main figures are not labeled with a number.

Reviewer #3 (Remarks to the Author):

Soya et al present an improved manuscript in this revision including additional experiments or modifications to the text that help and improve the work initially presented. I still miss some additional control experiments that they admit would be ideal but, due to limitation of time and/or animals, could not be performed. I agree that anxiety should be better measured by other paradigms

that are not specifically expected/demanded in this manuscript, as its focus is clearly fear, rather than anxiety. But it is good to keep in mind that anxiety is an unconditioned fear, and more comments or discussion would have been appreciated, in particular when discussing some results involving orexin activation and presenting to a novel context (e.g. fig7). In general, the manuscript has been improved with nice additional experiments and changes in the text that add clarity and help to make their claims stronger.

Responses to Reviewers

Reviewer #1:

Relatively minor concerns remain, including the choice of only one optogenetic stimulation frequency in LC (10 Hz) that is much higher than the spontaneous firing frequency in LC neurons (2-5 Hz) (Aston Jones et al., 1981). Also, Carter et al 2010 used optogenetic stimulation frequencies 1-10 Hz and frequencies >5 Hz appear to be saturating.

[Answer]

Thank you for your valuable suggestion. Because our optogenetic stimulation is applied on axon fibers, and not cell bodies, we do not think it is appropriate to compare these studies with ours. However, we agree that we should consider whether the condition of stimulation is appropriate. Actually, to set the condition, we referred many studies in which optogenetic excitation of NA neurons were performed, and works that showed firing rates of these cells. We had actually found many researchers used much higher stimulation frequencies than 1-10Hz (Jordan G McCall et al., 2015 Neuron, Gillian A Matthews et al., 2016 Cell, Lisa A Gunaydin et al., 2014 Cell, Tomonori Takeuchi et al., 2016 Nature). Takeuchi et al. (2016 Nature) reported that TH⁺ neurons in the LC showed burst firing at ranging from 15.3 to 28.2Hz when the mice were put into the novel cage, and they used 25 Hz soma stimulation of TH⁺ neurons in the LC for effective manipulation, although they also found the basal firing rate is 0.23-6.03 Hz. Based on these studies, we used 20 Hz stimulation in this study. We thought that our manipulation of NA fibers was sufficiently increase NA release in the LA, because Fos expression in the LA was strongly activated with this manipulation.

We added following sentence in the result section. “Although we did not monitor firing rate of LC^{NA} neurons in fearful condition, Takeuchi et al reported that these neurons showed burst firing at ranging from 15.3 to 28.2Hz when the mice were put into the novel cage, although basal firing rate is 0.23-6.03 Hz. Based on this observation, we used pulses of 473 nm illumination at 20 Hz (pulse width = 35 ms, interval = 15 ms, 10 mW) for the photostimulation.” (p.9, ll. 267-271).

“ Frequency of the stimulation was based on the previous study focusing on the firing rate of LC neurons (15.3 to 28.2 Hz) when the mice were put into the novel context⁴².” (p26, ll. 763-764)

Also, controls for CNO are necessary because this compound affects behavior under certain conditions and concentrations (Mc Claren et al., eNeuro 2016).

[Answer]

We agree that we should have excluded the possibility that CNO could elicit non-specific action on behavior. To address this, we injected CNO in WT mice 40 min before test session after cued fear conditioning (Fig. 2i), and found that 5 mg/kg CNO injection didn't show any effect on freezing response elicited with CS presentation, suggesting that we could exclude the non-specific effect of the CNO in this behavioral paradigm.

We added the sentences "To exclude the possibility that CNO elicits non-specific action on freezing behavior, we tested the effect of i.p. injection of CNO in wild type mice. After cued fear conditioning (Saline, $n = 5$; CNO, $n = 5$: Two-Way RM ANOVA, $F_{(1, 8)} = 0.03348$, $p = 0.8594$, Fig. 2h), fear response did not change between saline- and CNO injected group against cued (Two-Way RM ANOVA, $F_{(1, 8)} = 0.0003$, $p = 0.9847$) and contextual stimuli (Two-Way RM ANOVA, $F_{(1, 8)} = 0.00145$, $p = 0.9706$) (Figs. 2i-left, 2j-left). Average freezing showed no difference among these groups during CS presentation (unpaired two-tailed Student's t -test, $t = 0.3564$, $p = 0.7314$) or exposure to the context A (unpaired two-tailed Student's t -test, $t = 0.03794$, $p = 0.9706$) (Figs. 2i-right, 2j-right)." (p.5, ll. 142-150).

Reviewer #2:

The authors have done a moderate job of addressing reviewer concerns. Some of my concerns were addressed, however there are still two major technical concerns remaining that preclude publication in my opinion.

Multiple reviewers raised a concern about the lack of causal evidence that the orexin-LC-LA system is functional, since the authors only provide info about orexin-LC/NA and then LC/NA-LA. The authors respond that they "believe their work, albeit directly, clearly supports..." Do they mean indirectly? They don't show that the same neurons that are receiving orexin input in LC are the ones projecting to LA (in a functional manner, only with cTRIO).

[Answer]

To address the reviewer's suggestion, we examined Fos expression in the LA in fasted wild type mice with focal microinjection of an OX1R antagonist, SB334867 in the LC, because fasting is well-established to increase activity of orexin neurons. As we expected, Fos

expression in the LA was increased by fasting, and this activation was attenuated with blocking OX1R signaling in the LC by SB compound, suggesting that LA activity might be regulated with noradrenergic signaling by OX1R expressed in the LC. This pathway is involved in fasting induced freezing behavior in context A' (Fig.7). We believe this finding suggests that activation of orexin results in activation of the LA neurons via excitation of LC neurons through OX1R, showing that orexin-LC-LA system is functional.

We added and modified the sentences in the result section (Fig.7) (p10, ll. 304-326).

In response to point 3b raised by reviewer 1, the authors refuse to do the appropriate controls. While I might accept the argument that a control virus group need not have both saline or CNO (though this would be optimal), they MUST have a CNO control group. There have been a number of studies raising the issue,

including <https://www.ncbi.nlm.nih.gov/pmc/articles/PMC5089539/>.

These reports (that CNO on its own has effects on startle behavior) and alters catecholaminergic tone. This is a major confound that the authors must control for. I am shocked by their response to refuse to do such critical controls in the face of evidence suggesting nonspecific effects of CNO. Citing studies that failed to the appropriate controls (that were accepted/published before these new concerns were raised in the community) is not acceptable. To the journal/authors: Despite my request to label the figures in the first round, this request was ignored and main figures are not labeled with a number.

[Answer]

As we mention in the reply to R1, we injected CNO in WT mice 40 min before test session after cued fear conditioning (Fig. 2i). 5 mg/kg CNO injection didn't affect freezing elicited with CS presentation, suggesting that the CNO itself did not affect freezing behavior in this behavioral paradigm (Figs. 2h-j).

We added following sentences in the result section.

"To exclude the possibility that CNO elicits non-specific action on freezing behavior, we tested the effect of ip injection of CNO in wild type mice. After cued fear conditioning (Saline, $n = 5$; CNO, $n = 5$: Two-Way RM ANOVA, $F_{(1, 8)} = 0.03348$, $p = 0.8594$, Fig. 2h), fear response did not change between saline- and CNO injected group against cued (Two-Way RM ANOVA, $F_{(1, 8)} = 0.0003$, $p = 0.9847$) and contextual stimuli (Two-Way RM ANOVA, $F_{(1, 8)} = 0.00145$, $p = 0.9706$) (Figs. 2i-left, 2j-left). Average freezing showed no difference among these groups during CS presentation (unpaired two-tailed Student's t -test, $t = 0.3564$, $p = 0.7314$) or exposure to the context A (unpaired two-tailed Student's t -test, $t = 0.03794$, $p = 0.9706$) (Figs. 2i-right, 2j-right)." (p.5, ll. 142-150).

Reviewer #3

Soya et al present an improved manuscript in this revision including additional experiments or modifications to the text that help and improve the work initially presented. I still miss some additional control experiments that they admit would be ideal but, due to limitation of time and/or animals, could not be performed. I agree that anxiety should be better measured by other paradigms that are not specifically expected/demanded in this manuscript, as its focus is clearly fear, rather than anxiety. But it is good to keep in mind that anxiety is an unconditioned fear, and more comments or discussion would have been appreciated, in particular when discussing some results involving orexin activation and presenting to a novel context (e.g. fig7). In general, the manuscript has been improved with nice additional experiments and changes in the text that add clarity and help to make their claims stronger.

[Answer]

Thank you for the valuable comment. We understand the importance to consider the anxiety effect in our results. We added the sentences discussing about anxiety in 'Discussion' section.

We added the sentences "In other words, the increase in activity of $LH^{orexin} \rightarrow LC^{NA} \rightarrow LA$ circuits might be required for shifting anxiety state to the actual fear." (p14, ll. 429-431)

REVIEWERS' COMMENTS:

Reviewer #1 (Remarks to the Author):

The authors have adequately addressed the previous concerns.

Reviewer #2 (Remarks to the Author):

I have no further comments, and support publication.

Reviewer #3 (Remarks to the Author):

thank you for addressing my remaining concerns

Answers to REVIEWERS' COMMENTS:

Reviewer #1 (Remarks to the Author): The authors have adequately addressed the previous concerns.

>Thank you very much for your valuable input throughout the review process.

Reviewer #2 (Remarks to the Author): I have no further comments, and support publication.

>Thank you very much for your valuable input throughout the review process.

Reviewer #3 (Remarks to the Author): thank you for addressing my remaining concerns

>Thank you very much for your valuable input throughout the review process.